# Learning to See by Looking at Noise

**Manel Baradad**[*]
MIT CSAIL
mbaradad@mit.edu

**Jonas Wulff**[*]
MIT CSAIL
jwulff@csail.mit.edu

**Tongzhou Wang**
MIT CSAIL
tongzhou@mit.edu

**Phillip Isola**
MIT CSAIL
phillipi@mit.edu

**Antonio Torralba**
MIT CSAIL
torralba@mit.edu

## Abstract

Current vision systems are trained on huge datasets, and these datasets come with costs: curation is expensive, they inherit human biases, and there are concerns over privacy and usage rights. To counter these costs, interest has surged in learning from cheaper data sources, such as unlabeled images. In this paper, we go a step further and ask if we can do away with real image datasets entirely, by learning from procedural noise processes. We investigate a suite of image generation models that produce images from simple random processes. These are then used as training data for a visual representation learner with a contrastive loss. In particular, we study statistical image models, randomly initialized deep generative models, and procedural graphics models. Our findings show that it is important for the noise to capture certain structural properties of real data but that good performance can be achieved even with processes that are far from realistic. We also find that diversity is a key property for learning good representations.

## 1 Introduction

The importance of data in modern computer vision is hard to overstate. Time and again we have seen that better models are empowered by bigger data. The ImageNet dataset [1], with its 1.4 million labeled images, is widely thought to have spurred the era of deep learning, and since then the scale of vision datasets has been increasing at a rapid pace; current models are trained on up to one billion images [2]. In this paper, we question the necessity of such massive training sets of real images.

Instead, we investigate a suite of procedural noise models that generate images from simple random processes. These are then used as training data for a visual representation learner.

We identify two key properties that make for good synthetic data for training vision systems: 1) naturalism, 2) diversity. Interestingly, the most naturalistic data is not always the best, since naturalism can come at the cost of diversity. The fact that naturalistic data help may not be surprising, and it suggests that indeed, large-scale real data has value. However, we find that what is crucial is not that the data be *real* but that it be *naturalistic*, i.e. it must capture certain structural properties of real data. Many of these properties can be captured in simple noise models (Fig. 1).

The implications of our work are severalfold. First, our results call into question the true complexity of the problem of vision – if very short programs can generate and train a high-performing vision system, then vision may be simpler than we thought, and might not require huge data-driven systems to achieve adequate performance. Second, our methods open the door to training vision systems without reliance on datasets. The value in this is that datasets are encumbered by numerous costs:

---

[*]Equal contribution

35th Conference on Neural Information Processing Systems (NeurIPS 2021).

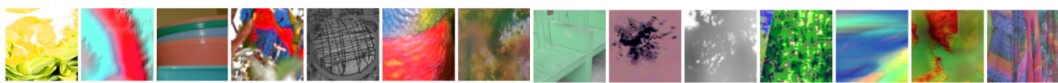

Figure 1: Which of these image crops come from real photos and which are noise? See footnote for answers[2]. These crops are examples of what MoCo v2 [3] sees as input in our experiments.

they may be expensive, biased, private, or simply intractable to collect. We do not argue for removing datasets from computer vision entirely (as real data might be required for evaluation), but rather reconsidering what can be done in their absence.

## 2    Related work

### 2.1    A short history of image generative models

Out of the $\mathbb{R}^{3n^2}$ dimensional space spanned by $3 \times n^2$ color images, natural images occupy a small part of that space, the rest is mostly filled by noise. During the last decades, researchers have studied the space of natural images to build models capable of compressing, denoising and generating images. The result of this research is a sequence of generative image models of increasing complexity narrowing down the space occupied by natural images within $\mathbb{R}^{3n^2}$.

One surprising finding is that natural images follow a power law on the magnitude of their Fourier transform [4, 5]. This is the basis of Wiener image denoising [6] and scale-invariant models of natural images [7, 8]. Dead leaves model [8, 9] was an attempt at generating images that explained the power law found in natural images and inspired the Retinex algorithm [10]. The multiscale and self-similar nature of natural images inspired the use of fractals [11, 12, 13] as image models.

Coding research for TV [5] and image modeling [6, 14, 15, 16] showed another remarkable property of natural images: the output values of zero mean wavelets to natural images are sparse and follow a generalized Laplacian distribution [6]. Color and intensity distributions in natural images have also been studied and found to follow rules that deviate from random noise [10, 17]. Research in texture synthesis showed how these statistical image models produced more realistic-looking textures [18, 19]. Those required fitting the image model parameters to specific images to sample more "like it". Recently, GANs [20] have shown remarkable image synthesis results [21]. Although GANs need real images to learn the network parameters, we show in this paper that they introduce a structural prior useful to encode image properties without requiring any training.

### 2.2    Training without real data and training with synthetic data

Through the above progression, generative models have become increasingly complex, with more parameters and more training data needed to fit these parameters. The same has happened with vision systems in general: state-of-the-art systems like BiT [22], CLIP [23], and SEER [2] obtain their best results on 300 million, 400 million, and 1 billion images respectively. These papers further show that such large data is critical to getting the best performance.

While this may be true, other work has shown that much smaller data is sufficient to already get decent performance. A single image can be enough to train, from scratch, a compelling generative model [24, 25] or visual representation [26] and, *even with no training data at all*, deep architectures already encode useful image priors that can be exploited for low-level vision tasks  [27] or for measuring perceptual similarity [28]. Our results, using an untrained StyleGANv2 [29] to generate training data, further affirm the utility of the structural priors in neural net architectures.

An alternative to training with real data is to train on synthetic data. This approach has been widely used in low-level tasks like depth, stereo, or optical flow estimation [30, 31, 32, 33], where 3D rendering engines can provide densely annotated data to learn from. Interestingly, for this class of tasks diversity is more important than realism [34], making procedurally generated scenes an effective alternative to renderings designed by professional 3D artists [35, 36].

---

[2]Answers: 1,3,4,5,6,8,14 are from ImageNet images.

Recent work has also investigated using deep generative models as a source of synthetic data to train classifiers [37, 38] and visual representations [39], or to generate synthetic annotated data for other downstream tasks [40, 41, 42]. However, these generative models are still fit to real image datasets and produce realistic-looking images as samples.

In this paper we push even further away from realism, generating synthetic data from simple noise processes. The closest prior work in this direction is the pioneering work of [43], which used automatically generated fractals to pre-train neural networks that converge faster than their randomly initialized counterparts. While they demonstrated that fractals can be effective for pre-training, there is still a large gap compared to pre-training on real data. We explore a much broader range of noise processes, including many classic models from the image coding and texture synthesis literature.

The use of randomized training data has also been explored under the heading of domain randomization [44], where 3D synthetic data is rendered under a variety of lighting conditions to transfer to real environments where the lighting may be unknown. Our approach can be viewed as an extreme form of domain randomization that does away with the simulation engine entirely: make the training data so diverse that a natural image will just look like a sample from the noise process.

There is some evidence that biology takes a similar approach during the prenatal development of the vision system. "Retinal waves" – spontaneous, semi-random activations of the retina – are thought to entrain edge detectors and other simple structures in the developing mammalian brain [45].

# 3 A progression of image generative models

Here we provide the details for the image models we will use in this paper. We test a suite of generative models of the form $g_\theta : \mathbf{z} \to \mathbf{x}$, where $\mathbf{z}$ are stochastic latent variables and $\mathbf{x}$ is an image. We will treat image generation as a hierarchical process in which first the parameters of a model, $\theta$, are sampled, and then the image is sampled given these parameters and stochastic noise. The parameters $\theta$ define properties of the distribution from which we will sample images, for example, the mean color of the image. The sampling process is as follows: $\theta \sim p(\theta)$, $\mathbf{z} \sim p(\mathbf{z})$, and $\mathbf{x} = g_\theta(\mathbf{z})$, which corresponds to sampling images from the distribution $p(\mathbf{x}, \theta) = p(\mathbf{x}|\theta)p(\theta)$. The parameters $\theta$ that define the model may be fit to real data or not. We will explore the case where the parameters are not fit to real data but instead sampled from simple prior distributions. Next, we describe the generative image models that we will evaluate in this paper (Fig. 2).

## 3.1 Procedural image models

The first class of models belong to the family of procedural image models. Procedural models are capable of generating very realistic-looking images in specific image domains. We include in this set also fractals, although they could make a class on their own.

**Fractals:** Fractals have been shown to capture geometric properties of elements found in nature [46]. Consequently, image models consisting of renderings of human-designed shapes with fractal structure [43] are likely to reproduce patterns found in natural images.

**CG:** Simulators and game engines rely on a mixture of human-driven design and procedural methods to generate environments simulating real-world properties such as illumination, 3D, and semantics. Here we include three CG models popular in computer vision with available datasets: CLEVR [47], DMLab [48] and MineRL [49].

## 3.2 Dead leaves image models

The second family of models is Dead leaves, one of the simplest image models. We consider simple shapes (leaves) like circles, triangles and squares, which are positioned uniformly at random in the image canvas until it is covered. To produce each shape, we circumscribe it in a circle with a radius following an exponential distribution of parameter $\lambda$. This procedure has been shown to produce images that have similar statistics to natural images, such as having a $1/|f|^\alpha$ power spectrum [8] and non-gaussian distribution of derivatives and wavelet coefficients [50].

In this study we will consider four dead leaves models: **Dead leaves - Squares:** Only uses squares axis-aligned. **Dead leaves - Oriented:** Squares are randomly rotated. **Dead leaves - Shapes:** Leaves

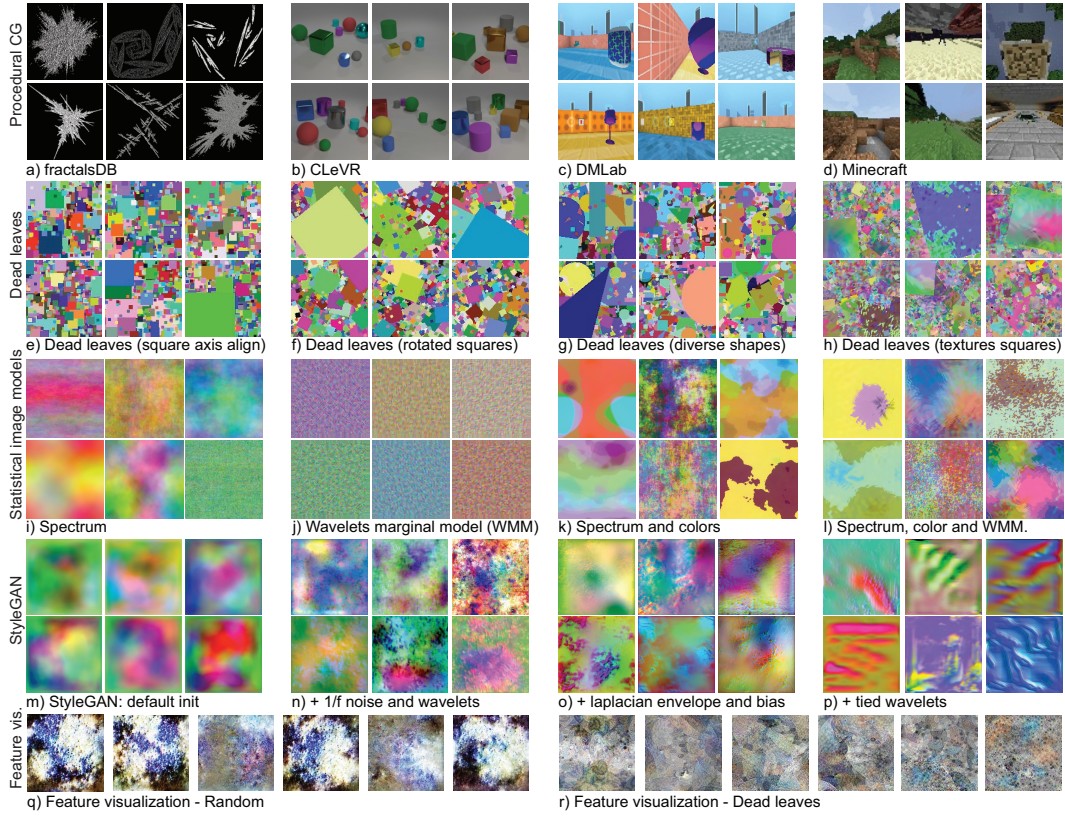

Figure 2: Samples from our synthetic datasets: a-d) CG synthetic images, e-h) Dead-leave models, i-l) Statistical image models, m-p) StyleGAN-based models with different weight initializations, and q-r) Feature visualizations.

can be circles, triangles and rectangles. **Dead leaves - Textured:** uses square leaves filled with a texture sampled from the statistical image models described in the next section.

### 3.3 Statistical image models

The third family of models is statistical image models with increasing complexity. Several generative models can be composed by using different combinations of properties.

**Spectrum:** The magnitude of the Fourier transform of many natural images follows a power law, $1/|f|^\alpha$, where $\alpha$ is a constant close to 1 [4]. In this generative model, we will sample random noise images constrained to have FT magnitude following $1/(|f_x|^a + |f_y|^b)$ with $a$ and $b$ being two random numbers uniformly sampled in the range $[0.5, 3.5]$. This model introduces a bias towards horizontal and vertical image orientations typical of natural images [51]. To generate color images we first sample three random orthogonal color directions and then generate power-law-noise on each channel independently. Samples from this model are shown in Fig. 2(i).

**Wavelet-marginal model (WMM):** Following [52], we generate textures by modeling their histograms of wavelet coefficients. To produce a texture, we create marginal histograms for the coefficients $c_i$ at $N$ scales ($i \in \{0...N-1\}$) and 4 orientations following a Generalized normal distribution centered at zero, thus $p(c_i) \propto \exp((-|c_i|/\alpha_i)^{\beta_i})$. Each scale $i$ represents the image down-scaled by a factor of $2^i$, and the parameters $\alpha_i$ and $\beta_i$ for each scale are $\alpha_i = 4^{2^i}$ and $\beta_i \sim 0.4 + \mathcal{U}(0, 0.4)$. In practice we use $N = 3$ and $N = 4$ for generating $128 \times 128$ and $256 \times 256$ resolution images respectively. Once we have sampled a marginal distribution of wavelet coefficients for each of the three channels, we do histogram matching iteratively starting from a Gaussian noise image following [53]. Fig. 2(j) shows samples from this model.

**Color histograms:** Here we take a generative model that follows the color distribution of the dead-leaves model. First we sample a number of regions $N \sim 3 + \lfloor \mathcal{U}(0, 20) \rfloor$, their relative sizes $S \sim 0.001 + \mathcal{U}(0, 1)$ and color at uniform. This results in a color distribution different from uniform.

Combining all these different models allows capturing color distributions, spectral components, and wavelet distributions that mimic those typical from natural images. Fig. 2(k) shows the result of sampling from a model that enforces random white noise to have the power-law spectrum and the color histogram according to this model. Fig. 2(l) shows samples from a model incorporating all of those properties (spectrum, color and WMM). Those models produce intriguing images but fail to capture the full richness of natural images as shown in Fig. 2(i-l).

## 3.4 Generative adversarial networks

The fourth family of models is based on the architecture of GANs. Commonly, the parameters of GANs are trained to generate realistic samples of a given training distribution. Here, *we do not use adversarial training or any training data.* Instead, we explore different types of initializations, study the usefulness of the GAN architecture as a prior for image generation, and show that effective data generators can be formed by sampling the model parameters from simple prior distributions. We use an untrained StyleGANv2 [29], and modify its initialization procedure to obtain images with different characteristics. This results in four classes of StyleGAN initializations:

**StyleGAN-Random** is the default initialization. Fig. 2(m) shows samples from this model. They lack high-frequency image content since the noise maps are not applied at initialization.

**StyleGAN-High-freq.** In this model, we increase high-frequency image content by sampling the noise maps as $1/f^\alpha$ noise with $\alpha \sim \mathcal{U}(0.5, 2)$, which models the statistics of natural images [4]. Additionally, the convolutional filters on all layers are randomly sampled from a bank of $3 \times 3$ wavelet filters, and each sampled wavelet is multiplied by a random amplitude $\sim \mathcal{N}(0, 1)$. Note that using Wavelets as spatial filters is a common practice when hand-designing networks [54, 55] and seems to well capture the underlying general structure of visual data. The samples in Fig. 2(n) show that this model generates high-frequency structures which are fairly uniformly distributed across the image.

**StyleGAN-Sparse.** Natural images exhibit a high degree of sparsity. In this model, we increase the sparsity of the images through two modifications. First, we modulate the $1/f^\alpha$ noise maps using a Laplacian envelope. We sample a $4 \times 4$ grid of i.i.d. Laplacian noise, resize it to the desired noise map resolution using bicubic upsampling and multiply this envelope with the original sampled $1/f^\alpha$ noise. Second, at each convolution, we add a random bias $\sim \mathcal{U}(-0.2, 0.2)$, which, in conjunction with the nonlinearities, further increases sparsity. Fig. 2(o) shows that the images created by this model indeed appear sparser. Yet, they are still lacking discernible image structures.

**StyleGAN-Oriented.** Oriented structures are a crucial component of natural images. We found that an effective way to introduce such structures to the previous models is to tie the wavelets, i.e. to use the same wavelet for all output channels. Under tied wavelets, the standard convolution becomes $y_k = \sum_l [a_{k,l}(x_l \star f_l)] + b_k$, where $y_k$ denotes output channel $k$, $x_l$ denotes input channel $l$, $b_k$ is a bias term, $a_{k,l} \sim \mathcal{N}(0, 1)$ is a random amplitude multiplier and the wavelet $f_l$ depends only on the input channel, but is shared across all output channels. As can be seen in Fig. 2(p), this creates visible, oriented structures in the output images.

## 3.5 Feature visualizations

The final class of models we study is feature visualizations [56]. CNN's can be used to produce novel images by optimizing the output of single or multiple units with respect to the input image, which is initialized with noise. Although these methods are commonly used to interpret and visualize the internal representation of a network, here we can use it as an image generative process. Following the procedure in [57], we obtain feature visualizations by optimizing the value of a single or a combination of two units of a neural network. We select those units from the layer that is typically used as feature extractor (i.e. the output of the penultimate linear layer), as we have empirically found this to yield more image-like visualizations compared to shallower layers. We create two datasets using this technique. **Feature vis. - Random:** ResNet50 with the default random initialization, shown in Fig. 2(q) and, 2) **Feature vis. - Dead leaves:** ResNet50 trained with dead leaves with diverse shapes, using MoCo v2 [3] and 1.3M sampled images. Samples are shown in Fig. 2(r).

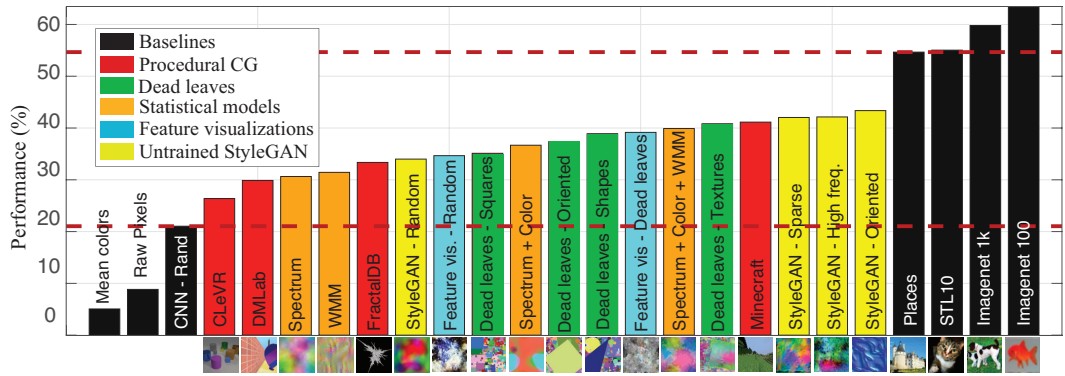

Figure 3: Top-1 accuracy for the different models proposed and baselines for Imagenet-100 [60]. The horizontal axis shows generative models sorted by performance. The two dashed lines represent approximated upper and lower bounds in performance that one can expect from a system trained from samples of a generic generative image model.

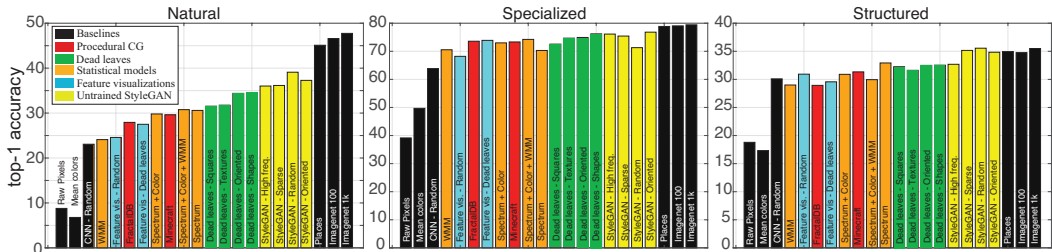

Figure 4: Average top-1 accuracy across the tasks in each of the three groups of tasks in VTAB [61]. The horizontal axis shows the different image generative models sorted by the average performance across all tasks. DMLab and CLEVR trained networks are not tested, as these datasets are present on some VTAB tasks.

## 4 Experiments

To study the proposed image generation processes, we train an AlexNet-based encoder using the Alignment and Uniformity loss proposed in [58], which is a form of contrastive loss theoretically equivalent to the popular InfoNCE loss [59]. We generate 105k samples using the proposed image models at $128 \times 128$ resolution, which are then downsampled to $96 \times 96$ and cropped at random to $64 \times 64$ before being fed to the encoder. After unsupervised training, we evaluate linear training performance (without finetuning) on the representation right before the projection layer, following standard practice [58, 59]. We fix a common set of hyperparameters for all the methods under test to the values found to perform well by the authors of [58]. Further details of the training are provided in the Sup.Mat.

We evaluate performance using Imagenet-100 [60] and the Visual Task Adaptation Benchmark [61]. VTAB consists of 19 classification tasks which are grouped into three categories: a) Natural, consisting of images of the world taken with consumer cameras b) Specialized, consisting in images of specialized domains, such as medical or aerial photography and c) Structured, where the classification tasks require understanding specific properties like shapes or distances. For each of the datasets in VTAB, we fix the number of training and validation samples to 20k at random for the datasets where there are more samples available.

As an upper-bound for the maximum expected performance with synthetic images, we consider the same training procedure but using the following real datasets: 1) Places365 [62] consisting of a wide set of classes, but a different domain 2) STL-10 [63], consisting of only 10 classes of natural images and 3) Imagenet1k [1], a superset of Imagenet100. As baselines we use mean image colors, raw pixels and features obtained by an untrained Alexnet (denoted CNN - Random).

| | Pretraining dataset | | | | | | | | |
|---|---|---|---|---|---|---|---|---|---|
| | Real images | | Procedural images | | | Other | | | |
| *Test* | I-1K | Places | CNN -Random | FractalDB | Dead leaves -Shapes | Feature Vis -Dead leaves | Spectrum +C+WMM | StyleGAN -Oriented | 4-Mixed |
| I-1K | 67.50 | 55.59 | 4.36 | 23.86 | 20.00 | 28.49 | 35.28 | 38.12 | 40.06 |
| I-100 | 86.12 | 76.00 | 10.84 | 44.06 | 38.34 | 49.48 | 56.04 | 58.70 | 60.66 |

Table 1: Performance of linear transfer for a ResNet50 pre-trained on different image models using MoCo-v2, on Imagenet-1K (I-1K) and Imagenet-100 (I-100). The 4-Mixed pretraining dataset corresponds to training with 4 of our datasets together, each with 1.3M images: Dead leaves - Shapes, Feature Vis. - Dead leaves, Spectrum + C + WMM and StyleGAN-Oriented.

## 4.1 Image model comparison

Figures 3 and 4 show the performance for the proposed fully generative methods from noise on Imagenet100 and VTAB (Tables can be found in the Sup.Mat.). The results on both datasets show an increased performance for Natural dataset (Imagenet100 and the Natural tasks in VTAB) that match the qualitative complexity and diversity of samples as seen in Fig. 2. On the other hand, Structured and Specialized tasks do not benefit as much from natural data (as seen in the middle and right-most plots in Fig. 4), and our models perform similarly for the tasks under test in this setting.

## 4.2 Large-scale experiments

Finally, we test one of the best-performing methods of each type on a large-scale experiment using a Resnet50 encoder instead of AlexNet. We generate 1.3M samples of the datasets at $256 \times 256$ resolution, and train using the procedure described in MoCo v2 [3] with the default hyperparameters for Imagenet1k (details of the training can be found in the Sup.Mat.).

The results in Table 1 show that the relative performance for the experiments with the Alexnet-based encoder is approximately preserved (except for dead leaves, which underperforms FractalDB-1k). Despite using the MoCo v2 hyperparameters found to be good for Imagenet1k (which may not be optimal for our datasets) and not using any real data, our best performing model achieves $38.12\%$ top-1 accuracy on linear classification on top of the learned features. Furthermore, we achieve $40.06\%$ top-1 accuracy by combining four of our datasets: Dead leaves - Shapes, Feature Visualizations - Dead leaves, Statistical (Spectrum + Color + WMM) and StyleGAN - Oriented.

Additionally, our image models allow training on arbitrary amounts of samples that can be generated on the fly. For StyleGAN - Oriented we found that training with the procedure described above but continuously sampling instead of using the fixed 1.3M images yields an improved top-1 accuracy of $38.94\%$.

## 5 What properties make for good generated data?

As we have seen, representations learned from noise-like images can be surprisingly powerful. Why is this the case? Intuitively, one might think that the features learned from these datasets themselves resemble noise. Interestingly, this is not the case. Fig. 5 shows the diversity of feature activations learned with the different datasets, extracted using the same procedure as described in Section 3.5. Though some datasets fail to capture certain properties (e.g. the Dead leaves - Squares dataset only reacts to axis-aligned images), feature visualizations qualitatively show that our datasets contain sufficient structure to learn a variety of complex features.

Yet, not all of our datasets reach the same performance, which raises the question of what makes a generated dataset good. Which properties are strongly correlated with performance? In the following, we investigate the statistical properties of individual images (5.1) and datasets as a whole (5.2)

## 5.1 Properties of individual images

Across our datasets, the image appearance varies significantly (see Fig. 2): They contain different structures and shapes, different color profiles, different levels of "realism", and different effects

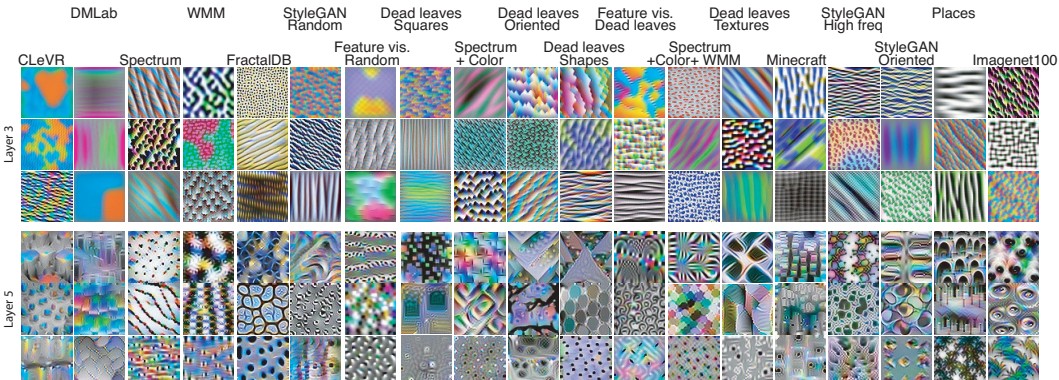

Figure 5: Feature visualizations for AlexNet-based encoder trained on some of the proposed datasets, for the 3rd and 5th (last) convolutional layer, using the procedure in [57].

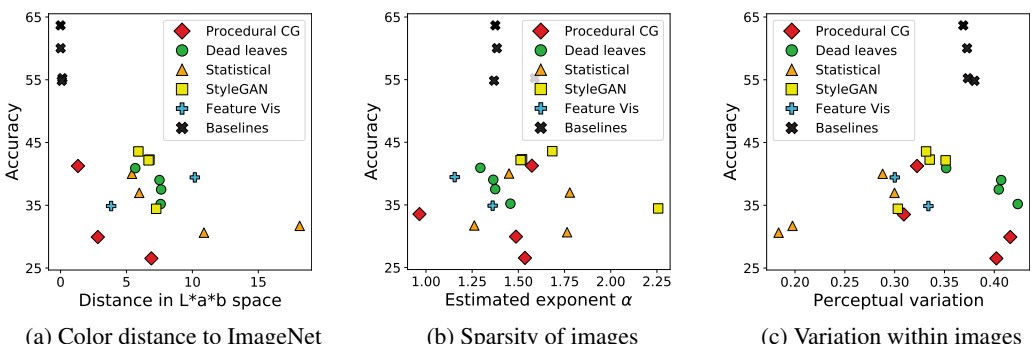

| (a) Color distance to ImageNet | (b) Sparsity of images | (c) Variation within images |

Figure 6: Different properties of the images have different impacts on accuracy. (a) Having a similar color distribution to natural images is important. (b) The spectral properties of dataset images are weakly correlated with accuracy, as long as they are within the right range. (c) There exists a sweet spot for variation within the images; having a higher variation results in less successful contrastive learning and decreased performance.

such as occlusions. How much of the difference in performance can be attributed to such low-level, per-image appearance characteristics?

**Color profile.** The first and most basic question is how important the color characteristics of our datasets are, since even in such a basic dimension as color, the datasets differ greatly (see Fig. 2). To test the impact of color characteristics on performance, we measure color similarity between the test dataset (here, ImageNet-100) and each of our pretraining datasets. First, we extract L*a*b values from 50K images from all datasets and ImageNet-100[3].The color distribution of each dataset can then be modeled as a three-dimensional Gaussian, and the color difference between two datasets can be computed as the symmetric KL divergence between those distributions. Fig. 6(a) shows that the color similarity of most of our datasets to natural images is fairly low; at the same time, we see a clear negative correlation between performance and color distance ($r = -0.57$).

**Image spectrum.** It is well known that the spectrum of natural images can be modeled as a heavy-tailed function $A/|f|^{\alpha}$, with $A$ a scaling factor and typically $\alpha \in [0.5, 2.0]$ [51]. How well do our datasets resemble these statistics, and how much does this impact performance? Fig. 6(b) shows that the statistics of most of our datasets fall within this range, except for Stylegan-default. However, while an exponent $\alpha \in [1.4, 1.7]$ seems to benefit performance, the correlation is weak.

**Image coherence.** Contrastive Learning utilizes different views as positive pairs during training, which are commonly generated using random augmentations of the same input image. For this to work, the images need a degree of global coherence, i.e. two views of the same image should be more

---

[3]In this and the next section, all measures were computed on 50K random images from the respective datasets, and, where applicable, compared against the reference statistics of ImageNet-100. To avoid distortions, correlations are always computed without taking the datasets ImageNet-100 and ImageNet1k into account.

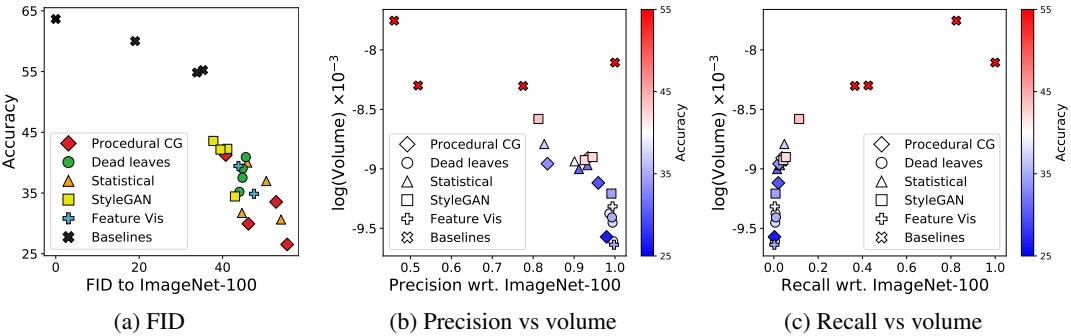

Figure 7: Properties of the datasets as a whole. (a) Resembling the test data is the best predictor of performance. Since a perfect overlap with the test data cannot be achieved, recall and precision are negatively correlated; (b) a high precision corresponds to datasets with small volume, resulting in low performance. (c) Recall is positively correlated with both volume and accuracy.

similar than two views from different images. To test the amount of coherence in our datasets and how this impacts downstream accuracy, we first compute the average perceptual variation within a dataset as $\frac{1}{N} \sum \text{LPIPS} \left( f \left( I_n \right), g \left( I_n \right) \right)$, where $f, g$ are two random crops of the same image and LPIPS is the perceptual distance from [28]. Fig. 6(c) shows the results. Interestingly, there seems to be a sweet spot of perceptual variation around $0.37$; for datasets with lower variation, perceptual variation is strongly correlated with accuracy ($r = 0.75$). Going beyond this sweet spot decreases accuracy, since now two augmentations of the same image are too dissimilar to provide good information. Finding a similar effect, [64] showed that there is a sweet spot for augmentation strength in contrastive learning.

## 5.2 Properties of the datasets

Beyond the properties of individual images, it is important to ask which properties of a dataset as a whole explain its level of performance. The first obvious predictor for good performance on a given test task is the distance of the training data distribution to the test data. We can quantify this using the Fréchet Inception Distance [65], which measures the distance between two sets of images by fitting Gaussians to the distribution of Inception features and measuring the difference between those Gaussians. As shown in Fig. 7(a), FID is strongly negatively correlated ($r = -.85$) with accuracy.

An ideal pretraining dataset would therefore resemble the test dataset as closely as possible, achieving a low FID as well as high precision and recall to the test dataset [66]. Interestingly, the trend we observe in our data points is in a different direction: Precision is *negatively* correlated with accuracy ($r = -0.67$), and the data shows a strong negative correlation between precision and recall ($r = -0.88$). This can be explained by the fact that our methods are not capable of perfectly reproducing the entirety of ImageNet, leaving two possible scenarios: Either a dataset is concentrated in a small part of the image manifold (images from CLEVR and DMLAB look naturalistic, yet are severely restricted in their diversity compared to natural images), or the dataset overshoots the manifold of test images, containing as much diversity as possible, in the limit encompassing the test dataset as a subset. Should a dataset be as realistic (i.e. maximize precision), or as diverse as possible?

An extreme version of the first case would be a training dataset that is concentrated around a few points of the test dataset, which would achieve high precision but low recall, and generally have low diversity. Given a distribution of features, we can measure the diversity of the dataset as $|\Sigma|$, i.e. the determinant of the covariance matrix in this feature space. Here, we use the same inception features as for the FID (see above). Fig. 7(b) shows precision vs. log-volume for all datasets; color indicates the performance on ImageNet-100. As can be seen, datasets with a high precision tend to not be particularly diverse and are negatively correlated with log-volume ($r = -0.84$). As volume increases, precision decreases, yet performance benefits.

Considering Recall, the picture changes. As shown in Fig. 7(c), higher recall is (after a certain point) positively correlated with both log-volume ($r = .79$) and accuracy ($r = .83$). Interestingly, this does *not* mean that precision is irrelevant – when controlling for recall, precision is again positively correlated with accuracy, albeit more weakly ($r = .22$). The reason for this behavior is that in case

of our datasets, precision and recall are negatively correlated, and thus present a trade-off. If such a trade-off is necessary when designing datasets for pretraining, our results indicate that maximizing recall might be more important than maximizing precision. Interestingly, despite a completely different setting, the same effect was observed in [34].

## 6  Conclusion

Can we learn visual representations without using real images? The datasets presented in this paper are composed of images with different types of structured noise. We have shown that, when designed using results from past research on natural image statistics, these datasets can successfully train visual representations. We hope that this paper will motivate the study of new generative models capable of producing structured noise achieving even higher performance when used in a diverse set of visual tasks. Would it be possible to match the performance obtained with ImageNet pretraining? Maybe in the absence of a large training set specific to a particular task, the best pre-training might not be using a standard real dataset such as ImageNet.

Although it might be tempting to believe that the datasets presented in this paper might reduce the impact of dataset bias, bias might still exist (although they might not come from social biases, they might still impact differently subsets of the test data) and it will be important to characterize it for each particular application.

**Acknowledgments:** Manel Baradad was supported by the LaCaixa Fellowship and Jonas Wulff was supported by a grant from Intel Corp. This research was partially conducted using computation resources from the Satori cluster donated by IBM to MIT.

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
