# Learning to See by Looking at Noise - Supplementary Material

## Contents

35th Conference on Neural Information Processing Systems (NeurIPS 2021).

## S-1  Experimental Details

### S-1.1  AlexNet-based encoder with Alignment and Uniformity loss

For the experiments in Section 4.1, we follow the training procedure described in [1], with parameters $\alpha = 2$, $t = 2$, the weight for both losses set to 1 and a batch size of 768. We train with stochastic gradient descent with momentum (set to 0.9) for 200 epochs, starting with a learning rate of 0.36 and decaying it by a factor of 0.1 at epochs 155, 170 and 185. The augmentations we use at train time are: transforming the image into gray-scale with probability 0.2; color, contrast and saturation jittering by a scaling factor sampled at uniform between 0.6 and 1.4; randomly flipping horizontally with probability 0.5; changing the aspect ratio by a factor between 0.75 and 1.33 sampled at uniform; and a random crop of the original image by a factor between 0.08 and 1 sampled at uniform. The dimensionality of the last and the penultimate embedding are 128 and 4096 respectively. The dimensionality of the third to last layer (used as the final representation) is 4096.

Finally, in Tables 1, 2 and 3 we report the detailed linear evaluation performance on Imagenet-100 and VTAB tasks with the previous setup and the proposed datasets.

### S-1.2  MoCo v2

For the experiments in Section 4.2, we follow the training procedure described in [2] with a ResNet-50 encoder. We use a batch size of 256 and the dimensionality of the last and the penultimate embedding are 128 and 4096 respectively. The rest of hyperparameters and image augmentations are the same as the ones found to work the best for Imagenet-1k in [2]. This corresponds to training for 200 epochs with a dataset of 1.3M samples, using a cosine learning rate scheduler starting at 0.015, temperature of 0.2 and the full set of augmentations described in [2].

| Model | I-100 | CIFAR | Flowers | Pets | SVHN | Caltech | DTD | Sun397 |
|---|---|---|---|---|---|---|---|---|
| **Baselines** | | | | | | | | |
| Pixels | 0.09 | 0.07 | 0.10 | 0.06 | 0.13 | 0.18 | 0.04 | 0.03 |
| Mean Colors | 0.05 | 0.06 | 0.05 | 0.04 | 0.20 | 0.03 | 0.06 | 0.04 |
| Random CNN | 0.21 | 0.22 | 0.22 | 0.14 | 0.38 | 0.38 | 0.16 | 0.11 |
| Places | 0.55 | 0.35 | 0.48 | 0.30 | 0.46 | 0.68 | 0.47 | 0.42 |
| Imagenet100 | 0.63 | 0.37 | 0.53 | 0.41 | 0.42 | 0.68 | 0.51 | 0.34 |
| Imagenet1k | 0.60 | 0.38 | 0.56 | 0.39 | 0.44 | 0.71 | 0.51 | 0.35 |
| **Statisticals** | | | | | | | | |
| S | 0.31 | 0.29 | 0.25 | 0.16 | 0.57 | 0.46 | 0.28 | 0.14 |
| WMM | 0.31 | 0.20 | 0.21 | 0.11 | 0.37 | 0.30 | 0.35 | 0.14 |
| S+C | 0.37 | 0.29 | 0.26 | 0.14 | 0.52 | 0.41 | 0.32 | 0.15 |
| S+C+WMM | 0.40 | 0.29 | 0.31 | 0.17 | 0.42 | 0.42 | 0.36 | 0.17 |
| **Feat. Visualizations** | | | | | | | | |
| Random | 0.35 | 0.22 | 0.23 | 0.14 | 0.36 | 0.36 | 0.28 | 0.13 |
| Dead leaves | 0.39 | 0.25 | 0.26 | 0.16 | 0.38 | 0.37 | 0.36 | 0.16 |
| **Procedurals** | | | | | | | | |
| FractalDB | 0.33 | 0.21 | 0.31 | 0.18 | 0.36 | 0.43 | 0.33 | 0.14 |
| Minecraft | 0.41 | 0.26 | 0.27 | 0.19 | 0.39 | 0.43 | 0.34 | 0.20 |
| **Dead leaves** | | | | | | | | |
| Squares | 0.35 | 0.25 | 0.31 | 0.18 | 0.42 | 0.53 | 0.32 | 0.20 |
| Oriented | 0.37 | 0.30 | 0.31 | 0.19 | 0.49 | 0.55 | 0.35 | 0.22 |
| Shapes | 0.39 | 0.28 | 0.34 | 0.20 | 0.45 | 0.56 | 0.35 | **0.23** |
| Textures | 0.41 | 0.25 | 0.34 | 0.22 | 0.37 | 0.50 | 0.32 | 0.22 |
| **StyleGANs** | | | | | | | | |
| Random | 0.34 | **0.40** | 0.38 | **0.22** | **0.67** | **0.57** | 0.31 | 0.18 |
| Sparse | 0.40 | 0.32 | 0.33 | 0.19 | 0.52 | 0.48 | 0.37 | 0.20 |
| High freq. | 0.42 | 0.32 | **0.42** | 0.21 | 0.46 | 0.50 | **0.41** | 0.21 |
| Oriented | **0.43** | 0.35 | 0.39 | 0.19 | 0.52 | 0.54 | 0.40 | 0.22 |

Table 1: Imagenet100 and VTAB linear evaluation results for Natural tasks (columns) after contrastive training on each of the datasets (rows). From left to right the columns correspond to the tasks: Imagenet100, CIFAR-100, Oxford Flowers102, Oxford IIIT Pets, SVHN, Caltech101, DTD and Sun397. In bold, best synthetic dataset, underlined when it also outperforms training with real images.

| Model | EuroSAT  | Resisc45  | Retino.  | Camelyon  |
|---|---|---|---|---|
| Baselines | | | | |
| Pixels | 0.18 | 0.12 | 0.74 | 0.52 |
| Mean Colors | 0.40 | 0.15 | 0.74 | 0.70 |
| Random CNN | 0.69 | 0.38 | 0.74 | 0.74 |
| Places | 0.88 | 0.71 | 0.74 | 0.82 |
| Imagenet100 | 0.90 | 0.72 | 0.74 | 0.80 |
| Imagenet1k | 0.90 | 0.73 | 0.74 | 0.80 |
| Statisticals | | | | |
| S | 0.82 | 0.49 | 0.74 | 0.77 |
| WMM | 0.81 | 0.51 | 0.74 | 0.77 |
| S+C | 0.85 | 0.56 | 0.74 | 0.78 |
| S+C+WMM | 0.85 | 0.60 | 0.74 | 0.79 |
| Feat. Visualizations | | | | |
| Random | 0.75 | 0.49 | 0.74 | 0.76 |
| Dead leaves | 0.83 | 0.58 | 0.74 | **0.81** |
| Procedurals | | | | |
| FractalDB | 0.83 | 0.59 | 0.74 | 0.79 |
| Minecraft | 0.84 | 0.57 | 0.74 | 0.78 |
| Dead leaves | | | | |
| Squares | 0.82 | 0.58 | 0.74 | 0.76 |
| Oriented | 0.86 | 0.63 | 0.74 | 0.77 |
| Shapes | 0.86 | 0.66 | 0.74 | 0.79 |
| Textures | 0.83 | 0.64 | 0.74 | 0.77 |
| StyleGANs | | | | |
| Random | 0.85 | 0.53 | 0.74 | 0.74 |
| Sparse | 0.86 | 0.61 | 0.74 | 0.79 |
| High freq. | 0.87 | **0.67** | **0.74** | 0.77 |
| Oriented | **0.88** | 0.64 | 0.74 | 0.81 |

Table 2: VTAB linear evaluation results for Specialized tasks (columns) after contrastive training on each of the datasets (rows). From left to right the columns correspond to the tasks: EuroSAT, Resisc45, Diabetic Retinopathy and Patch Camelyon. In bold, best synthetic dataset, underlined when it also outperforms training with real images.

| Model | ClevrD | ClevrC | dSprO | dSprL | sNorbE | sNorbA | DMLab | KittiD |
|---|---|---|---|---|---|---|---|---|
| **Baselines** | | | | | | | | |
| Pixels | 0.34 | 0.23 | 0.05 | 0.09 | 0.14 | 0.07 | 0.17 | 0.40 |
| Mean Colors | 0.27 | 0.24 | 0.07 | 0.06 | 0.13 | 0.05 | 0.24 | 0.33 |
| Random CNN | 0.49 | 0.34 | 0.17 | 0.29 | 0.22 | 0.13 | 0.34 | 0.42 |
| Places | 0.46 | 0.50 | 0.21 | 0.16 | 0.31 | 0.22 | 0.40 | 0.52 |
| Imagenet100 | 0.46 | 0.50 | 0.21 | 0.16 | 0.35 | 0.23 | 0.38 | 0.50 |
| Imagenet1k | 0.46 | 0.53 | 0.20 | 0.15 | 0.36 | 0.25 | 0.38 | 0.51 |
| **Statisticals** | | | | | | | | |
| S | 0.52 | 0.43 | 0.23 | 0.24 | 0.32 | 0.19 | 0.34 | 0.37 |
| WMM | 0.39 | 0.44 | 0.20 | 0.15 | 0.23 | 0.18 | 0.33 | 0.40 |
| S+C | 0.50 | 0.43 | 0.18 | **0.26** | 0.30 | 0.16 | 0.32 | 0.33 |
| S+C+WMM | 0.49 | 0.41 | 0.17 | 0.25 | 0.27 | 0.16 | 0.33 | 0.33 |
| **Feat. Visualizations** | | | | | | | | |
| Random | 0.49 | 0.38 | 0.18 | 0.25 | 0.27 | 0.13 | 0.34 | 0.43 |
| Dead leaves | 0.45 | 0.40 | 0.17 | 0.24 | 0.24 | 0.14 | 0.32 | 0.40 |
| **Procedurals** | | | | | | | | |
| FractalDB | 0.44 | 0.47 | 0.16 | 0.20 | 0.26 | 0.14 | 0.33 | 0.34 |
| Minecraft | 0.45 | 0.42 | 0.18 | 0.17 | 0.27 | 0.17 | 0.35 | **0.50** |
| **Dead leaves** | | | | | | | | |
| Squares | 0.51 | 0.48 | 0.18 | 0.16 | **0.35** | 0.18 | **0.38** | 0.32 |
| Oriented | 0.48 | 0.49 | 0.18 | 0.17 | 0.30 | 0.22 | 0.37 | 0.39 |
| Shapes | 0.48 | 0.51 | 0.17 | 0.19 | 0.32 | 0.21 | 0.38 | 0.36 |
| Textures | 0.49 | 0.48 | 0.19 | 0.24 | 0.27 | 0.17 | 0.38 | 0.32 |
| **StyleGANs** | | | | | | | | |
| Random | 0.51 | 0.45 | **_0.23_** | 0.24 | 0.34 | **0.24** | 0.38 | 0.46 |
| Sparse | **_0.53_** | 0.48 | 0.21 | 0.24 | 0.35 | 0.20 | 0.37 | 0.39 |
| High freq. | 0.47 | 0.45 | 0.20 | 0.23 | 0.33 | 0.19 | 0.35 | 0.38 |
| Oriented | 0.48 | **0.53** | 0.21 | 0.22 | 0.33 | 0.21 | 0.37 | 0.44 |

Table 3: VTAB linear evaluation results for Structured tasks (columns) after contrastive training on each of the datasets (rows). From left to right the columns correspond to the tasks: Clevr-Closest Object Distance, Clevr-Count, dSprites-Orientation, dSprites-Label X-position, SmallNORB-Elevation, sNORB-Azimuth, DMLab and KITTI-Closest Vehicle Distance. In bold, best synthetic dataset, underlined when it also outperforms training with real images.

# S-2 Analysis of datasets

Section 5.1 describes several comparisons between our synthetic datasets and real image datasets; in particular, we compare the color distribution, the sparsity / spectral magnitude, and self-similarity. Here, we present additional data for these experiments, and provide the full distributions for these criteria and all datasets.

## S-2.1 Color histograms

In the first experiment in Section 5.1, we compare the color distributions in L*a*b space of our synthetic datasets to real image datasets, in particular to ImageNet. We found that the color distribution of better performing datasets is closer to that of ImageNet; at the same time, the color distributions of all our datasets are still fairly different from ImageNet.

Figure 1 shows the color distributions for all datasets. For each dataset, we show two scatterplots: *L-vs-A*, to show the lightness properties of a given dataset, and *A-vs-B* to show the color distribution. Figure 1 confirms the differences between our datasets and ImageNet. In particular, our datasets lack both very dark and very bright areas; at the same time, the colors are too saturated. Interestingly,

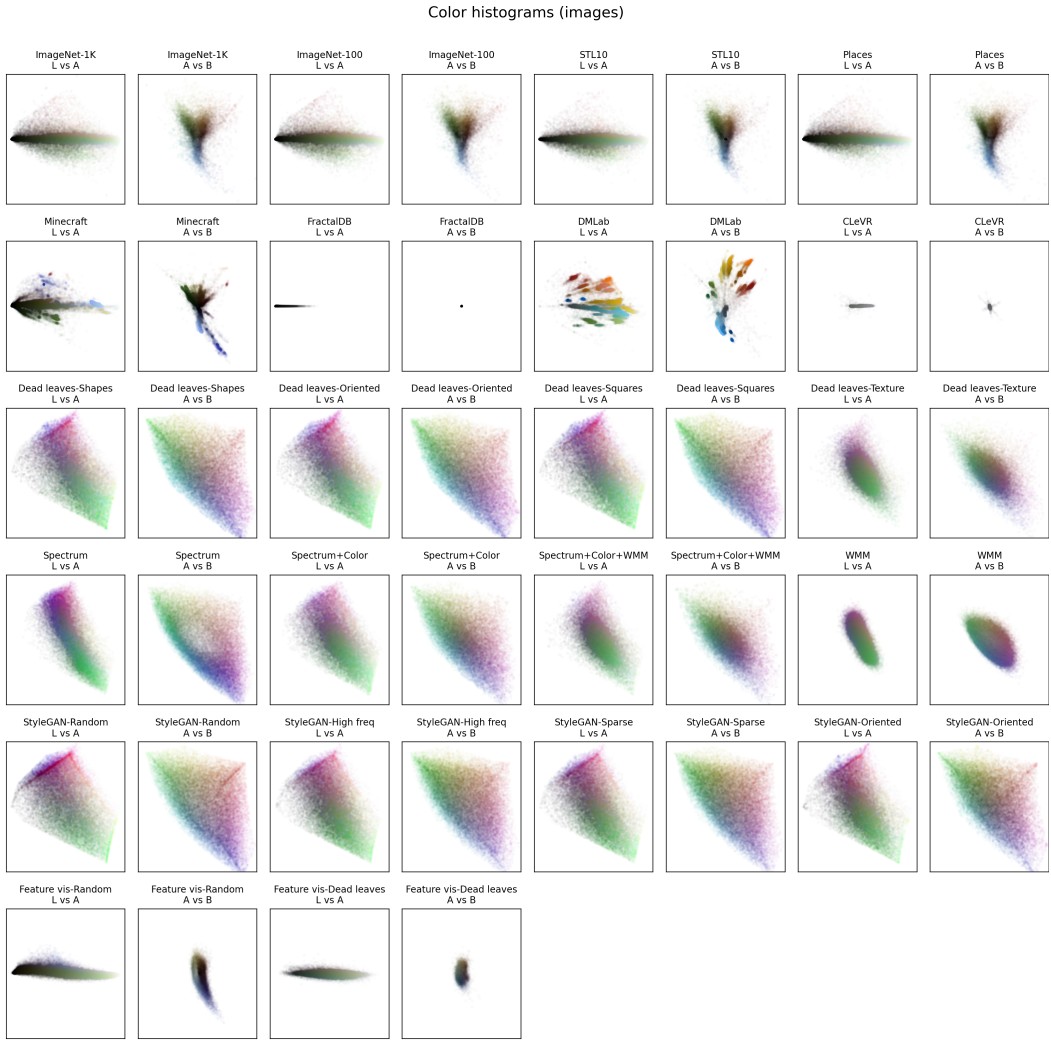

Figure 1: Color distributions for all datasets in L*a*b space. For each dataset, we plot the data in the L-A plane and in the A-B plane. The ranges are the same for all plots ($L \in [0, 100]$; $a, b \in [-100, 100]$).

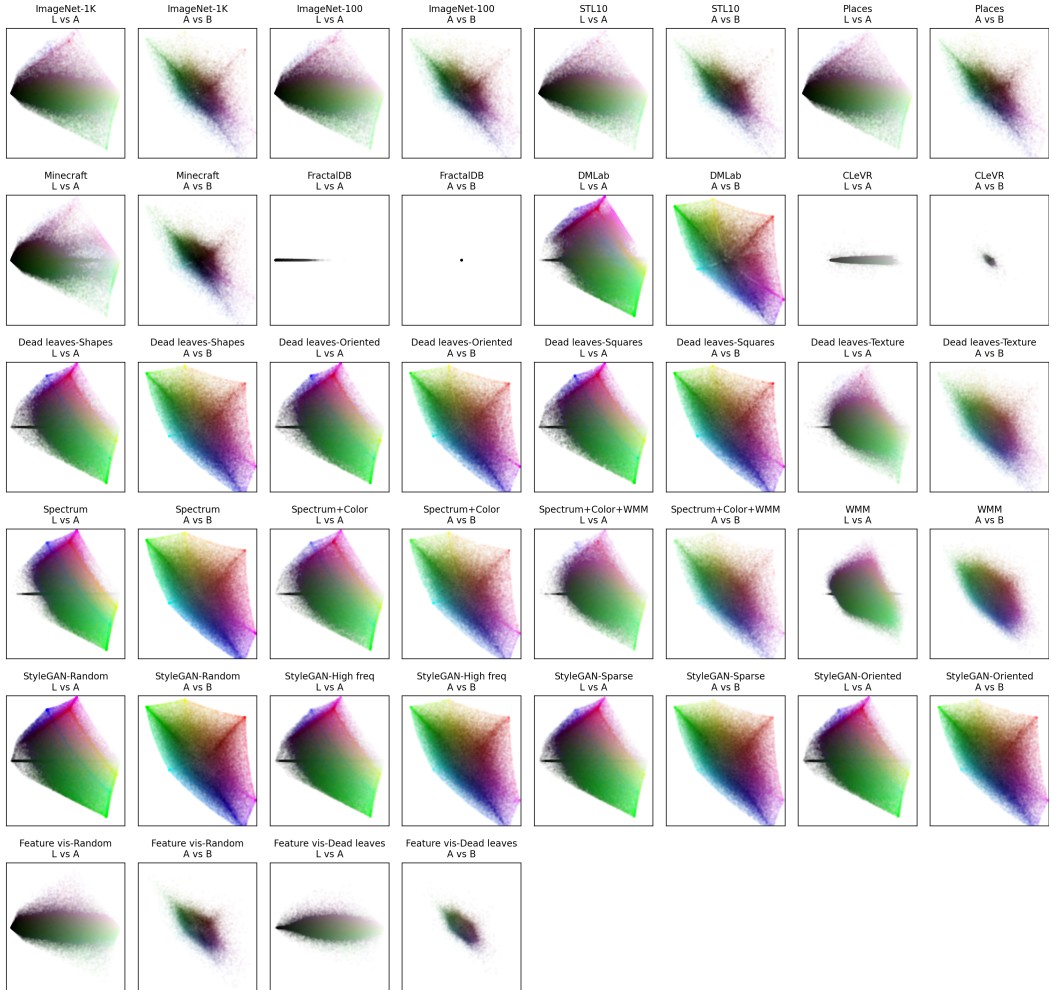

Figure 2: Color distributions *after augmentation*. For each dataset, we plot the data in the L-A plane and in the A-B plane. The ranges are the same for all plots ( $L \in [0, 100]; a, b \in [-100, 100]$).

except for *Minecraft*, existing datasets from computer graphics (second row) fare even worse, and either do not contain any colors at all (*FractalDB*), show a strange and distinctly clustered color distribution (*DMLab*), or are dominated by gray tones (*CLeVR*). We hypothesize that this is one of the main reasons why these datasets perform significantly worse than ours.

Figure 2 shows the color distribution after the MoCo-v2 augmentations. In all cases color jittering causes the colors to spread out more and saturation to increase. Interestingly, the synthetic datasets now contain a large number of dark but completely desaturated pixels. However, they still lack dark colors, and the oversaturation persists.

## S-2.2 Sparsity

In the second experiment in Section 5.1, we evaluate the sparsity characteristics of the image spectrum, in particular, whether the average spectral magnitude of images from the datasets follows the same well-known $\frac{1}{|f|^\alpha}$ pattern. While there seems to be some sweet spot around $\alpha = 1.35$, we did not find a particularly strong correlation between $\alpha$ and the achieved accuracy.

Figure 3 shows the full distribution of $\alpha$ values for each dataset. We find that the distributions of $\alpha$ are quite varied among datasets; interestingly, this does not seem to have a huge impact on performance.

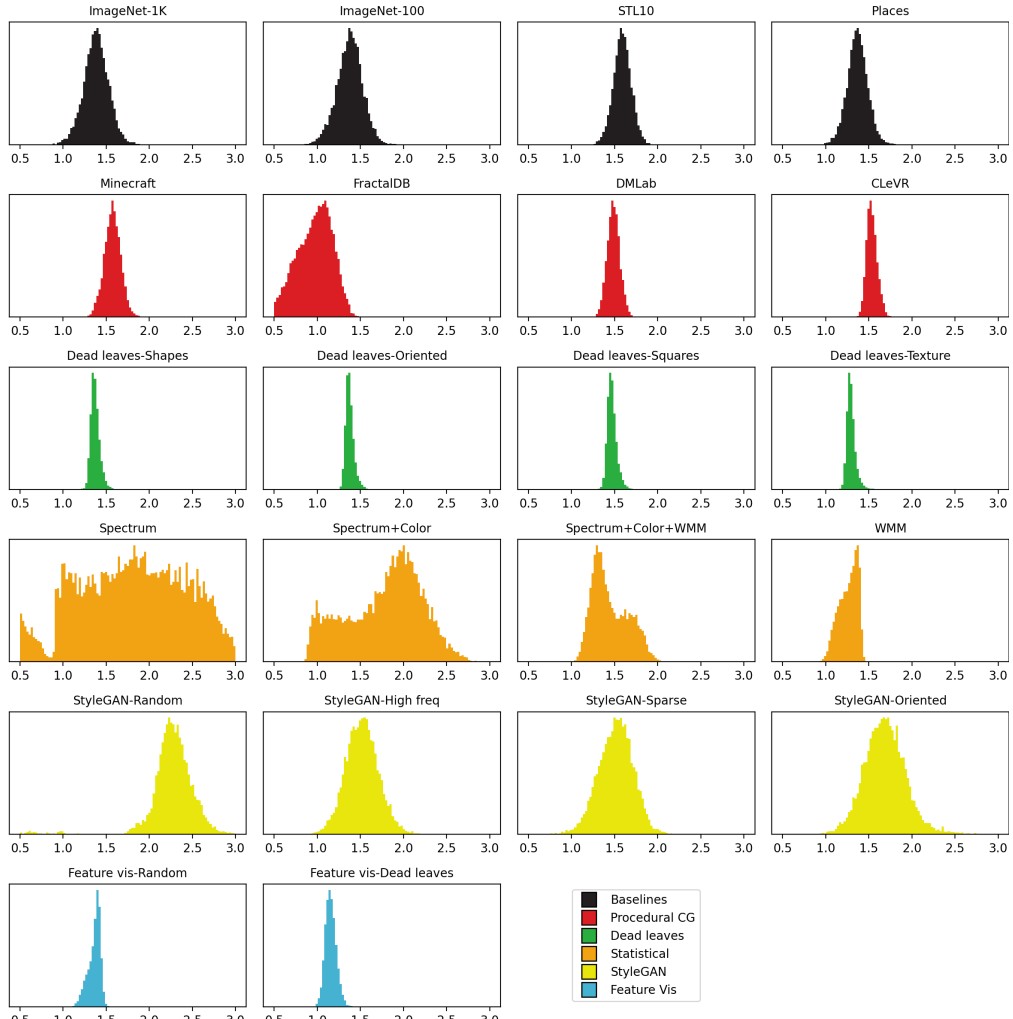

Figure 3: Distributions of $\alpha$ for different datasets. As in the main paper, color indicates the class of the datasets.

### S-2.3   Self-similarity

In the last experiment of Section 5.1, we compare self-similarity within images, which is a core requirement for contrastive learning. We measure self-similarity as the average perceptual distance between two crops of the same image, and find that there is an optimal value for self-similarity, up to which self-similarity and accuracy are strongly correlated, but after which this correlation becomes negative.

Figure 4 shows the full distributions of self-similarity values for all datasets; we find that the best performing datasets (*StyleGAN-Oriented, StyleGAN-High-freq, StyleGAN-Sparse, Minecraft*) also seem to be those for which the full distribution of self-similarity scores best resembles that of ImageNet.

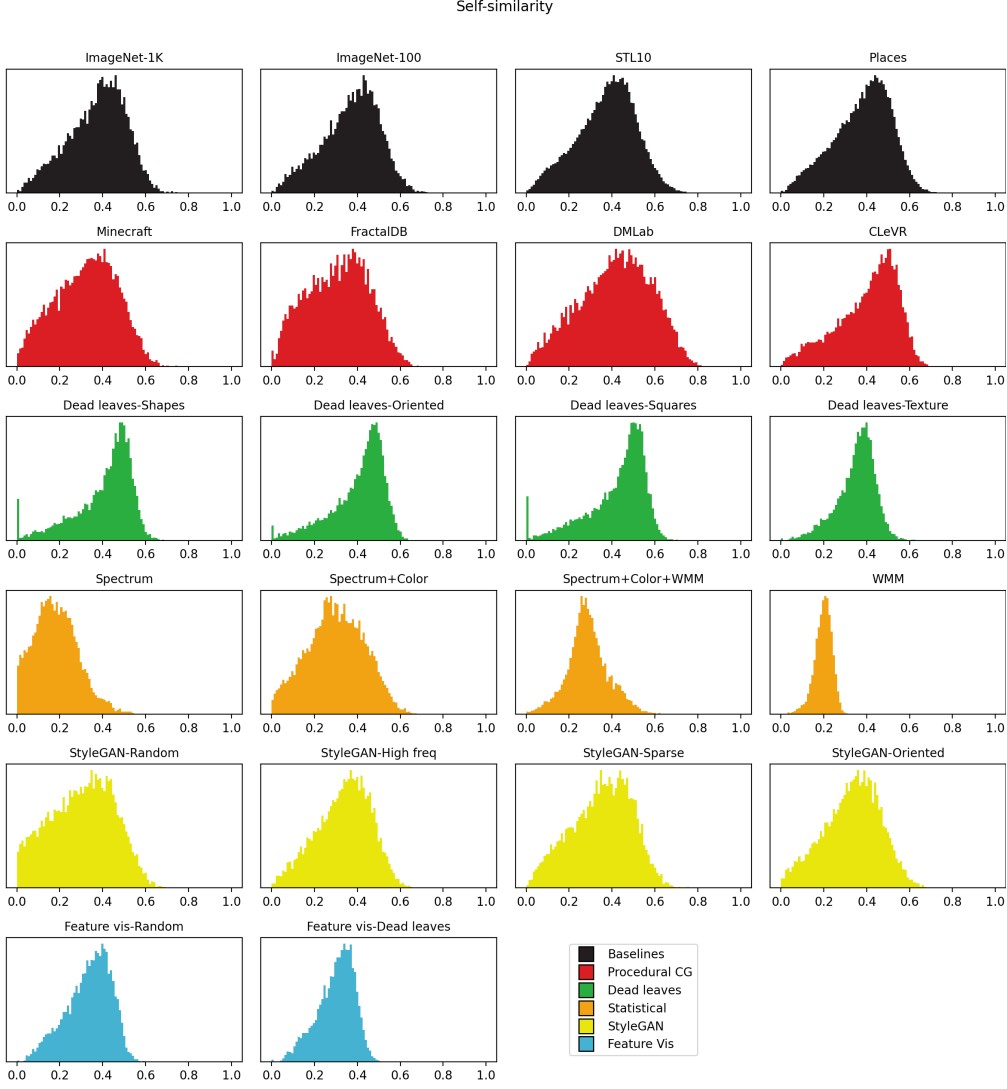

Figure 4: Distributions of self-similarity scores for different datasets, as computed using a perceptual similarity between different crops of the same image. As in the main paper, color indicates the class of the datasets.

# S-3    Dataset samples

## S-3.1    FractalDB

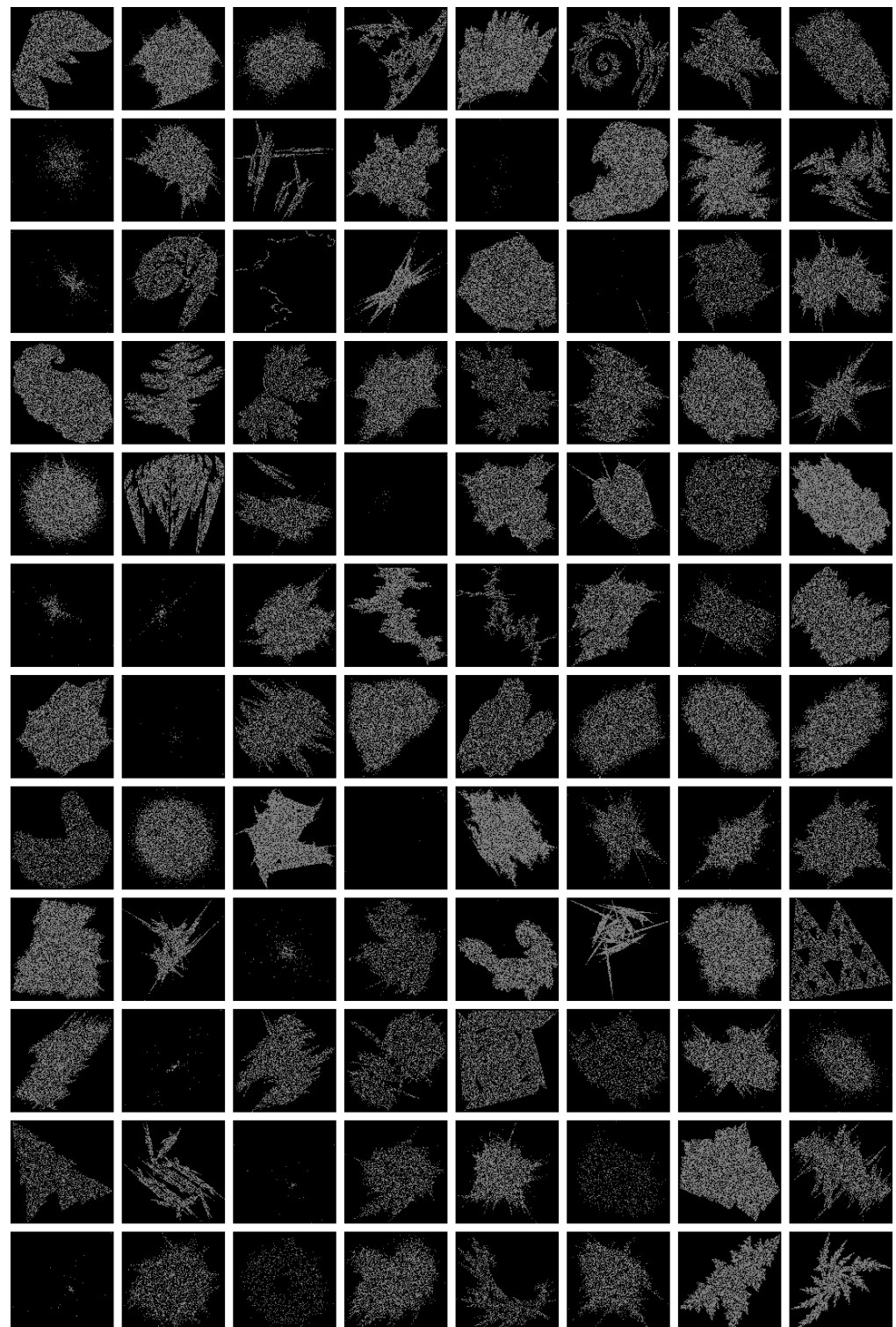

Figure 5: 96 random samples of the dataset a) FractalDB (letter as referenced in Fig. 2 of the main paper).

**S-3.2    CLeVR**

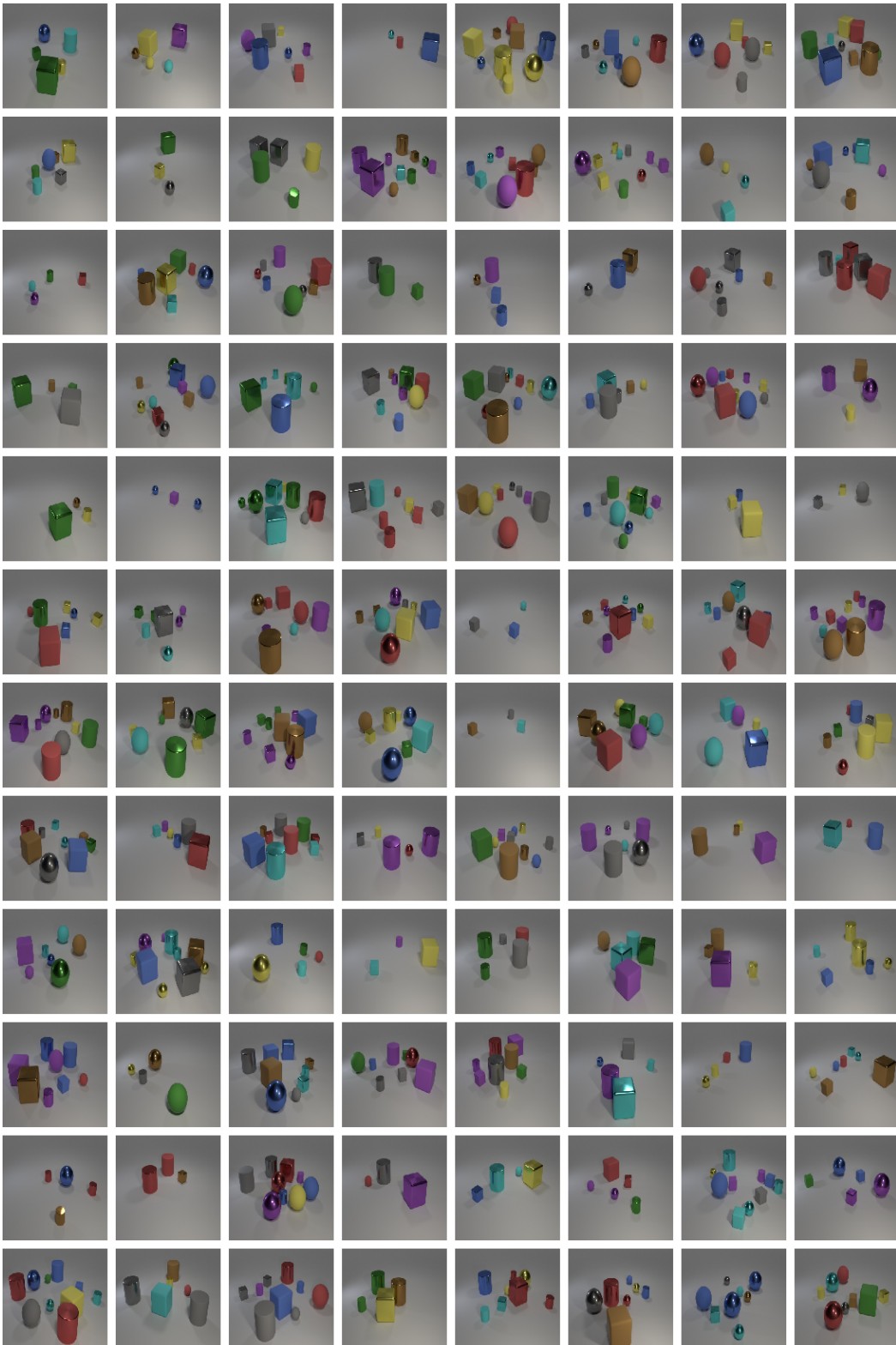

Figure 6: 96 random samples of the dataset b) CLeVR (letter as referenced in Fig. 2 of the main paper).

**S-3.3 DMLab**

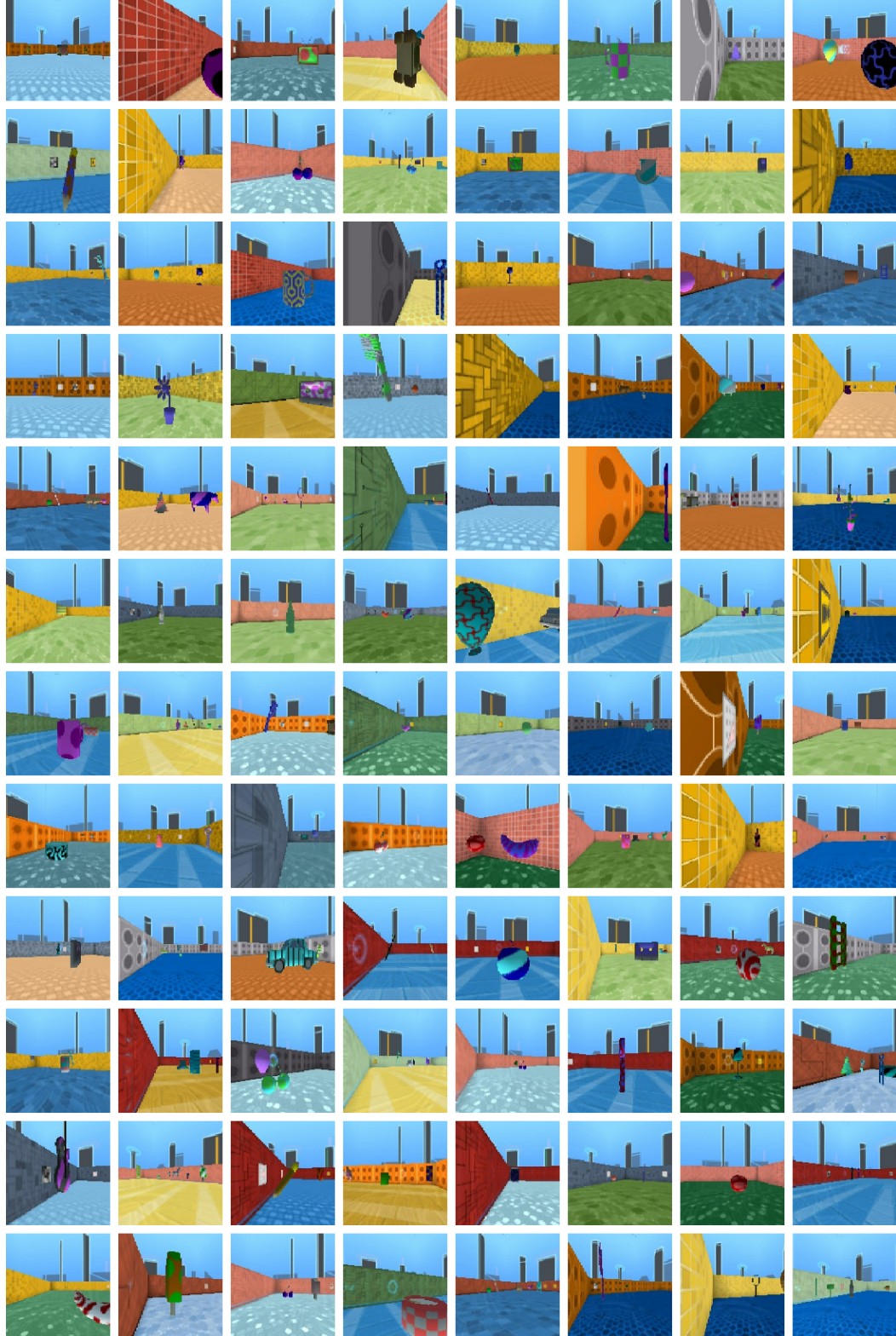

Figure 7: 96 random samples of the dataset c) DMLab (letter as referenced in Fig. 2 of the main paper).

## S-3.4 Minecraft

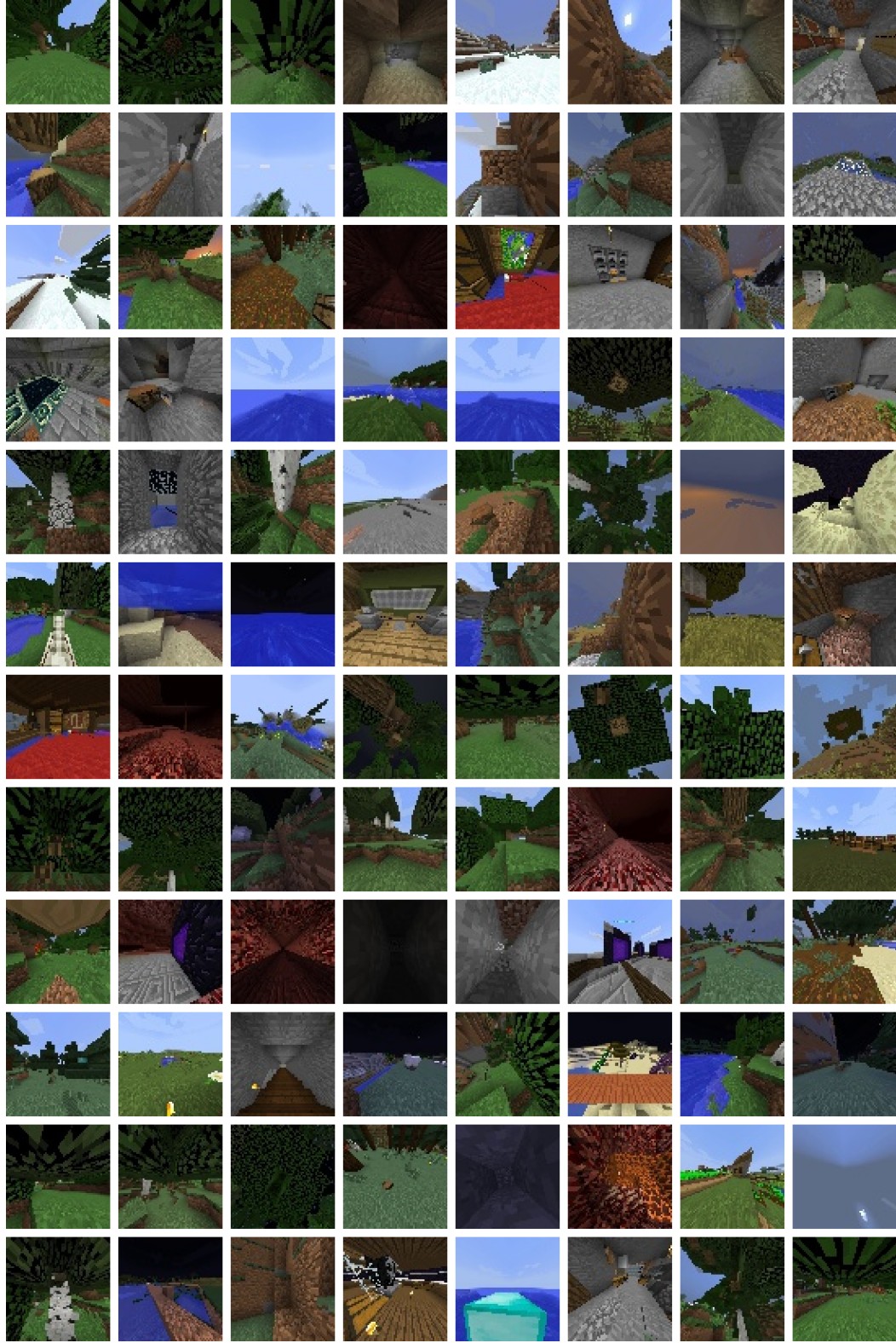

Figure 8: 96 random samples of the dataset d) Minecraft (letter as referenced in Fig. 2 of the main paper).

## S-3.5  Dead leaves - Squares

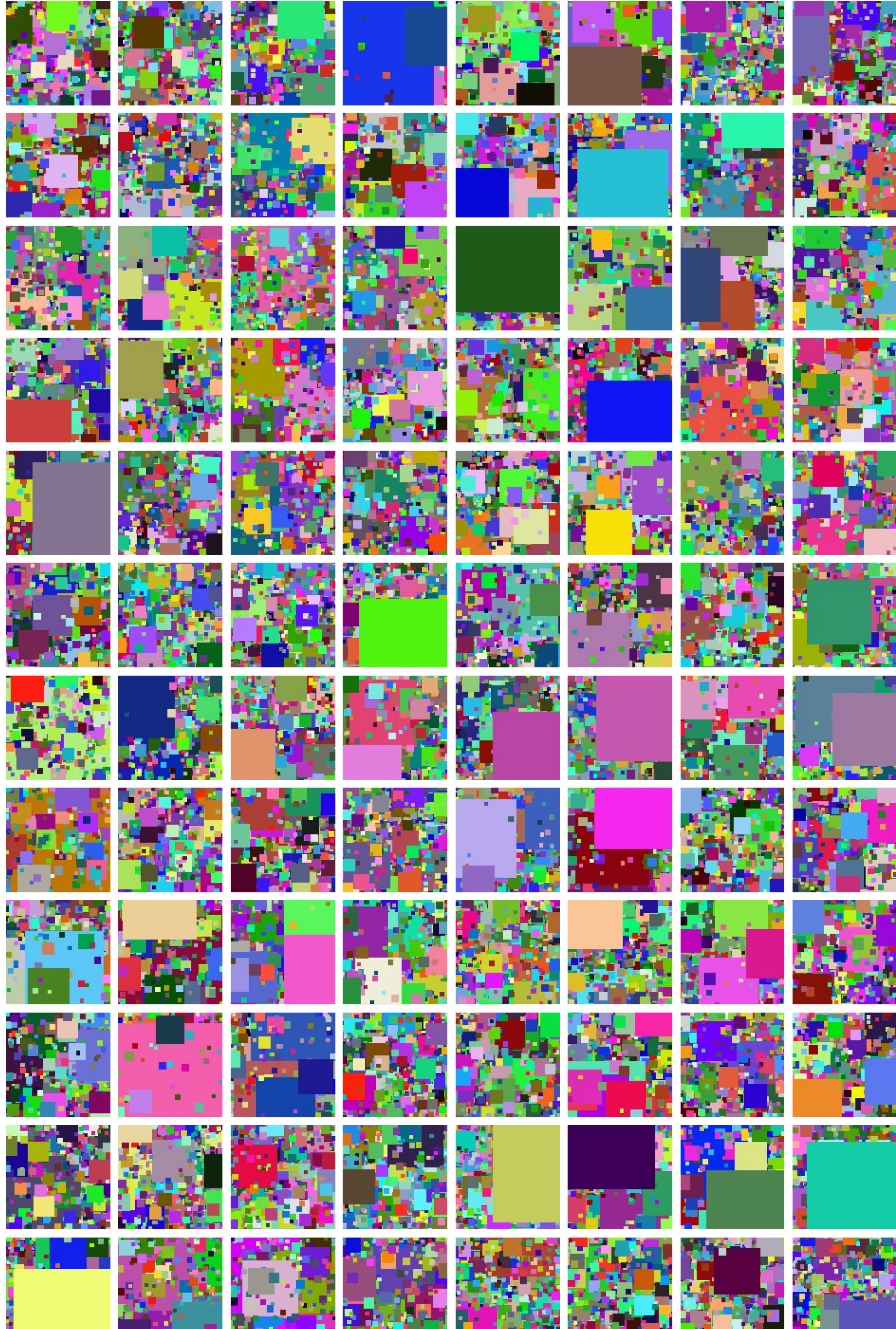

Figure 9: 96 random samples of the dataset e) Dead leaves - Squares (letter as referenced in Fig. 2 of the main paper).

## S-3.6  Dead leaves - Oriented

Figure 10: 96 random samples of the dataset f) Dead leaves - Oriented (letter as referenced in Fig. 2 of the main paper).

## S-3.7 Dead leaves - Shapes

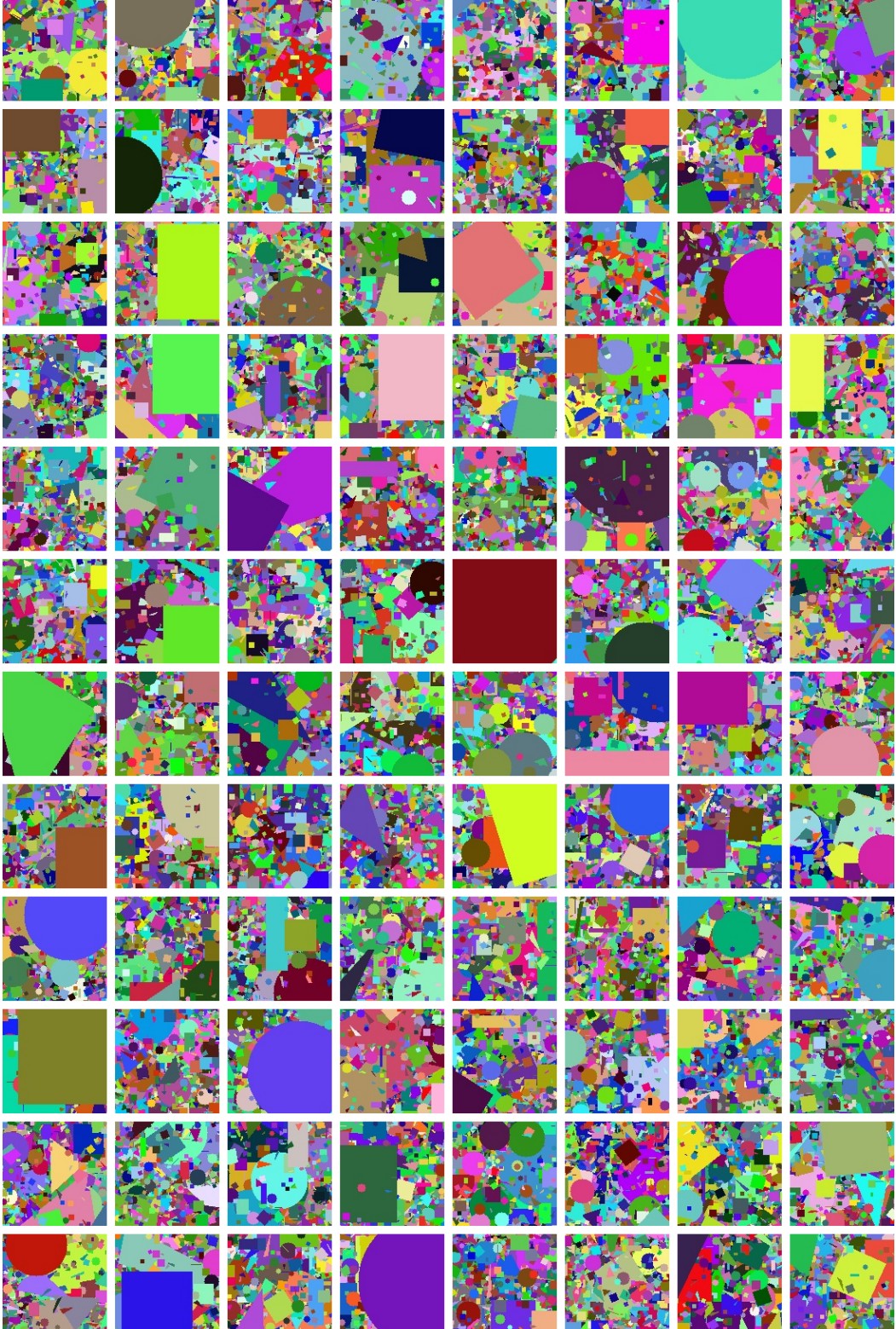

Figure 11: 96 random samples of the dataset g) Dead leaves - Shapes (letter as referenced in Fig. 2 of the main paper).

## S-3.8 Dead leaves - Textures

Figure 12: 96 random samples of the dataset h) Dead leaves - Textures (letter as referenced in Fig. 2 of the main paper).

## S-3.9 Spectrum

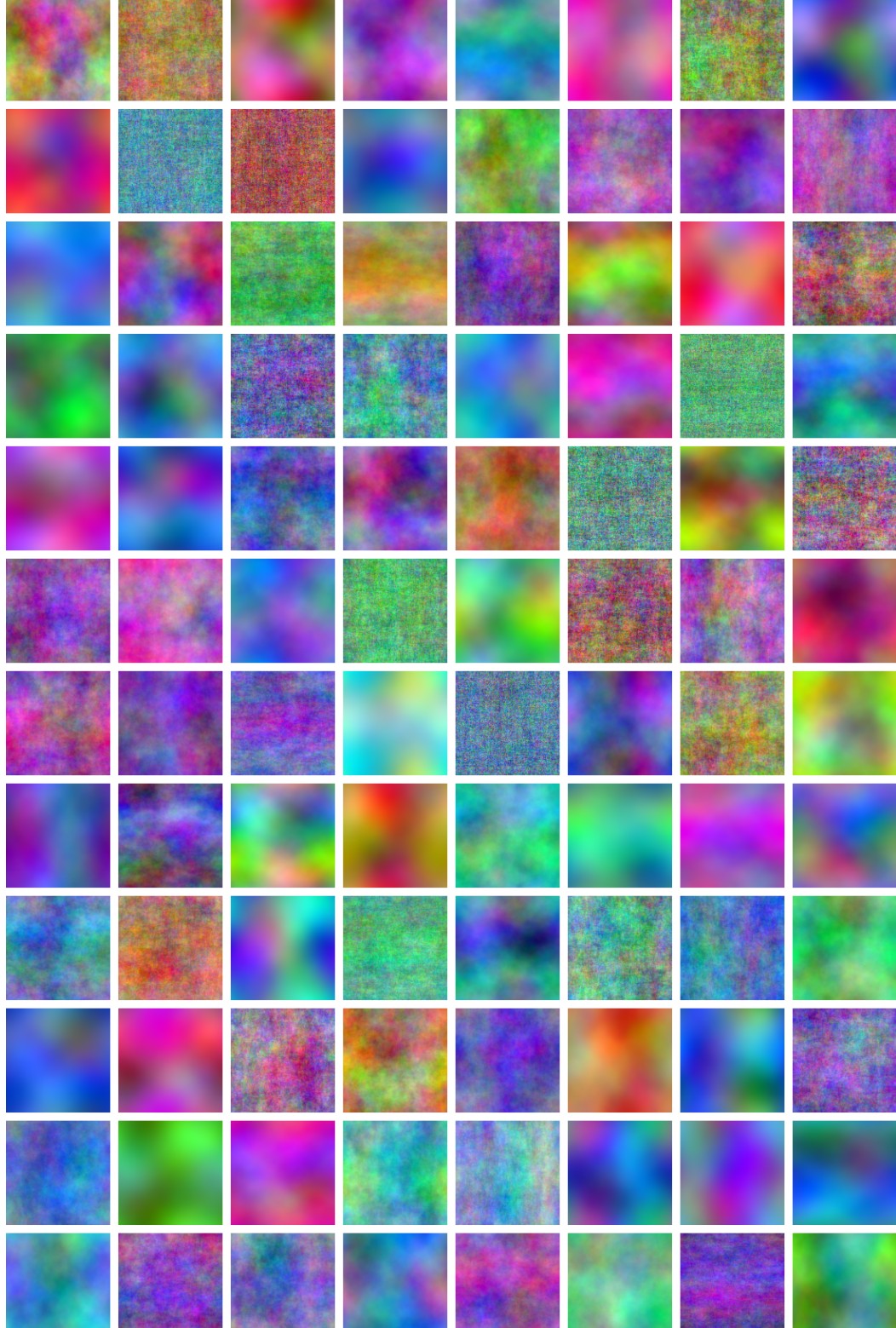

Figure 13: 96 random samples of the dataset i) Spectrum (letter as referenced in Fig. 2 of the main paper).

**S-3.10    WMM**

Figure 14: 96 random samples of the dataset j) WMM (letter as referenced in Fig. 2 of the main paper).

**S-3.11    Spectrum + Color**

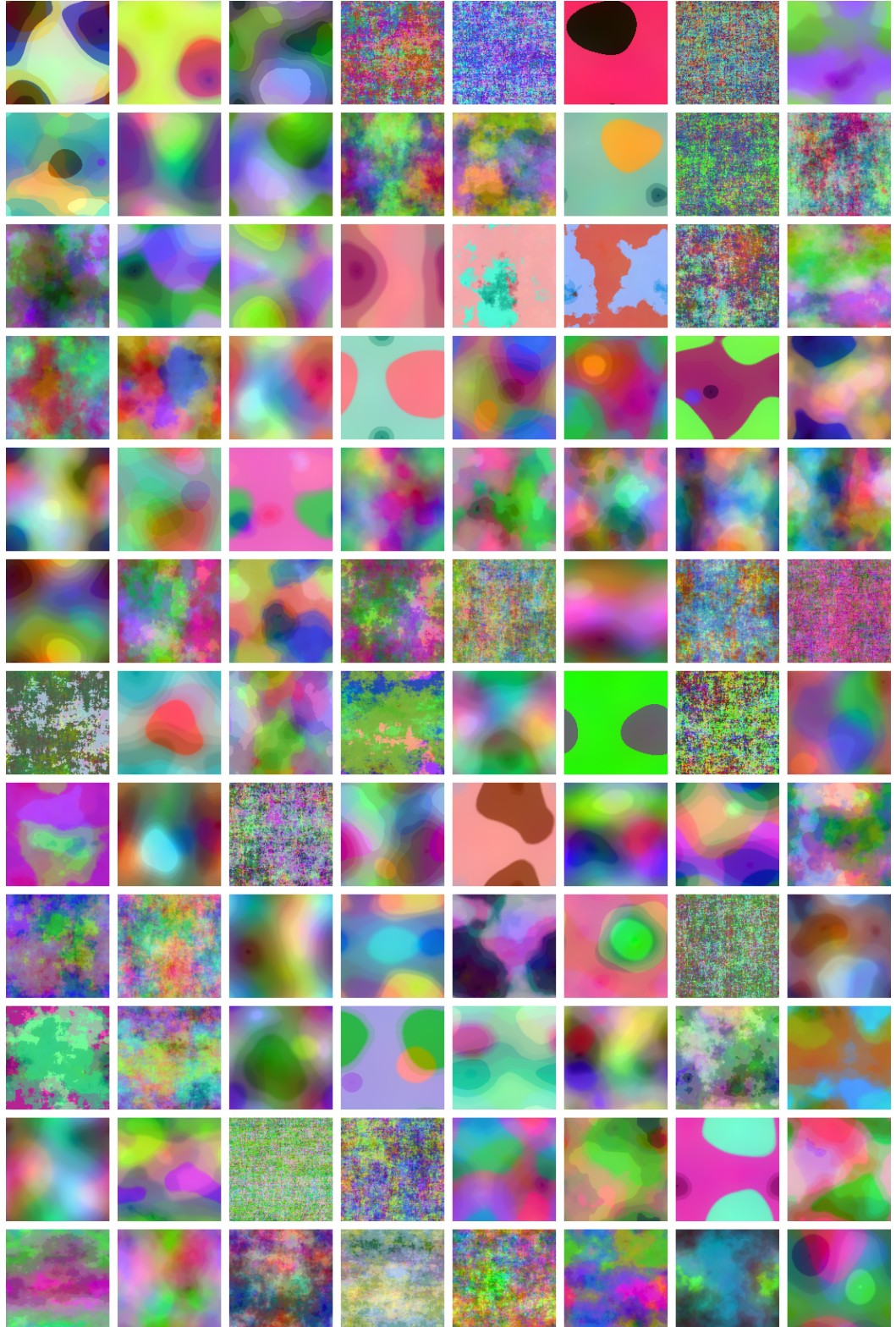

Figure 15: 96 random samples of the dataset k) Spectrum + Color (letter as referenced in Fig. 2 of the main paper).

## S-3.12 Spectrum + Color + WMM

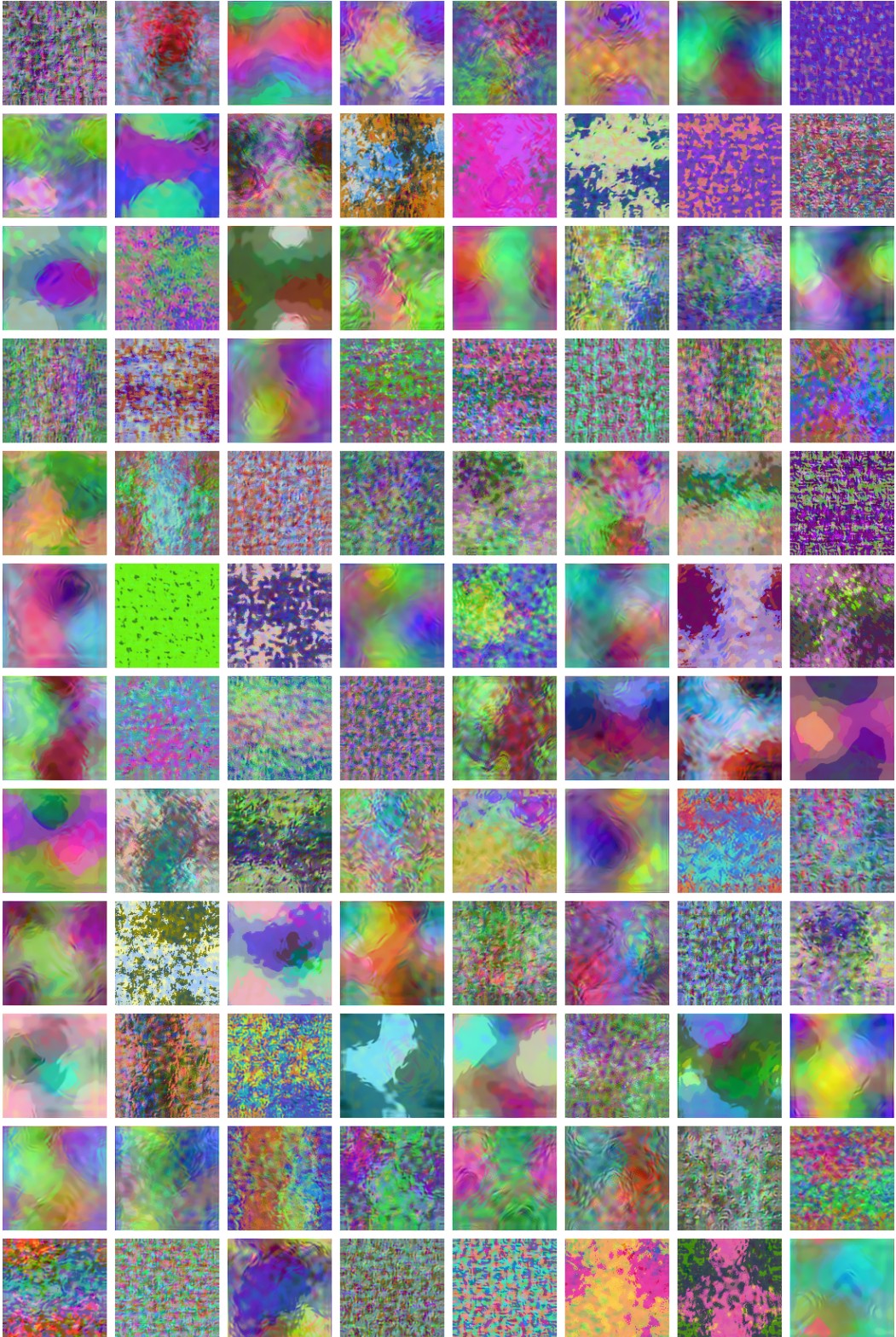

Figure 16: 96 random samples of the dataset l) Spectrum + Color + WMM (letter as referenced in Fig. 2 of the main paper).

**S-3.13 StyleGAN - Random**

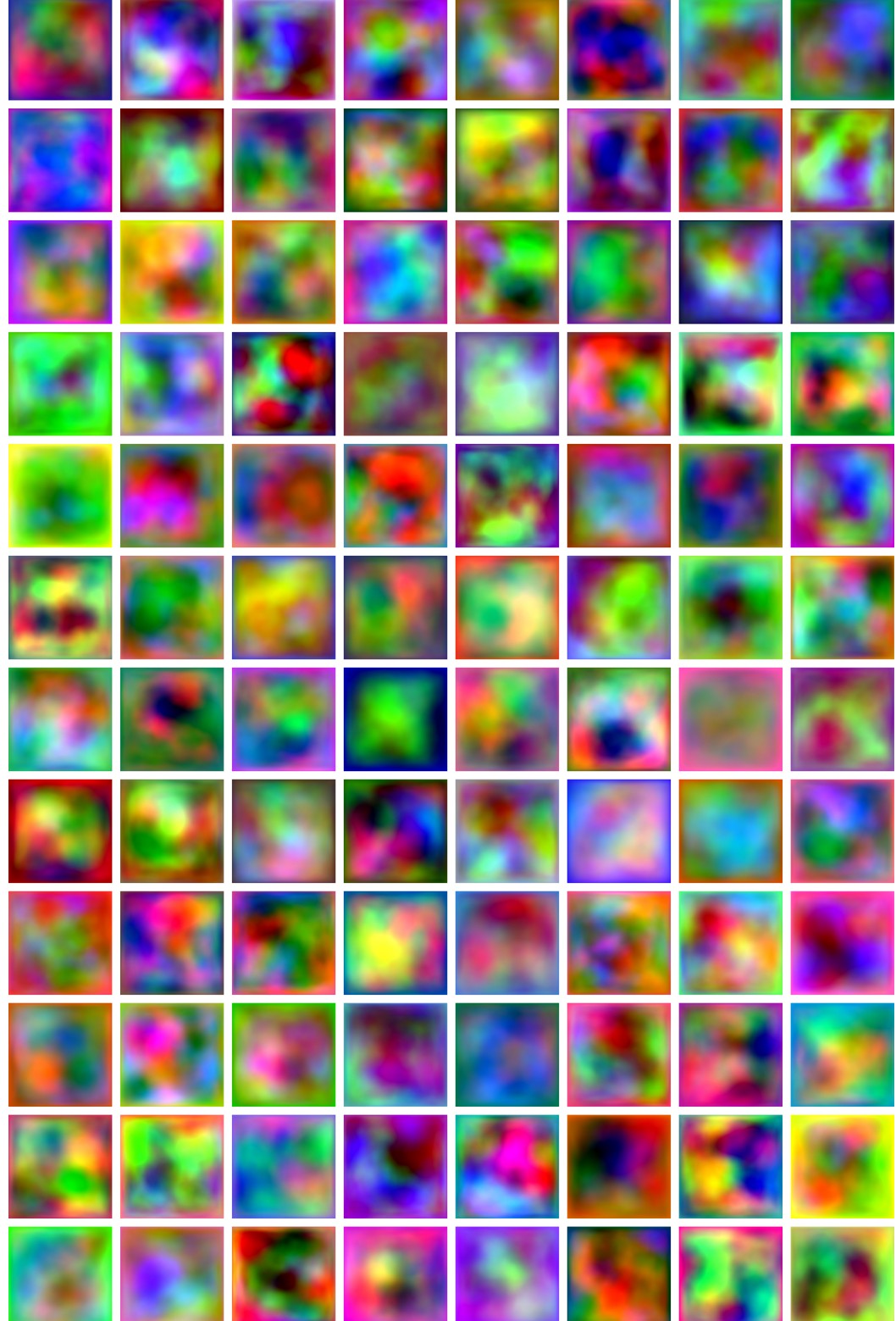

Figure 17: 96 random samples of the dataset m) StyleGAN - Random (letter as referenced in Fig. 2 of the main paper).

**S-3.14    StyleGAN - High freq.**

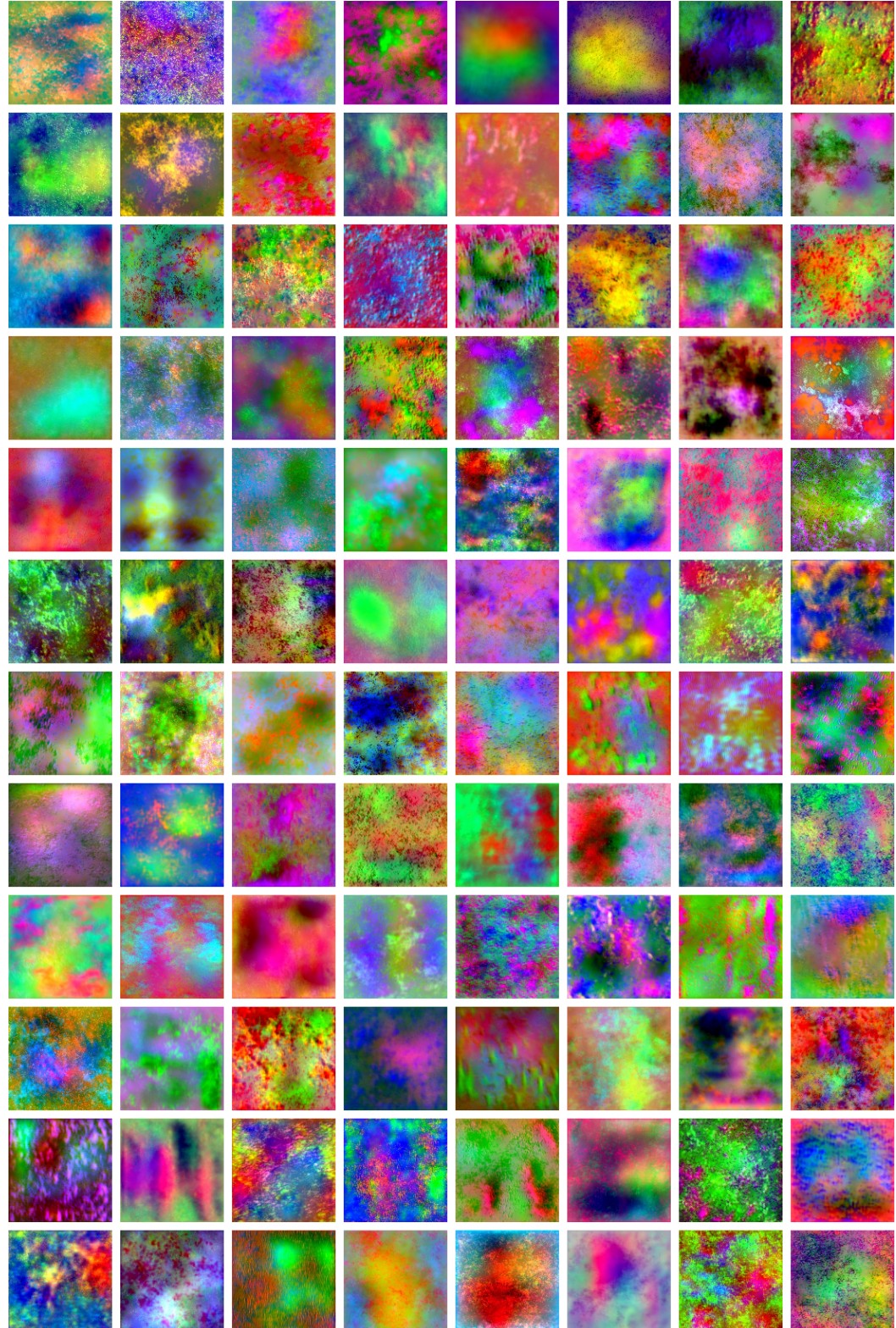

Figure 18: 96 random samples of the dataset n) StyleGAN - High freq. (letter as referenced in Fig. 2 of the main paper).

## S-3.15  StyleGAN - Sparse

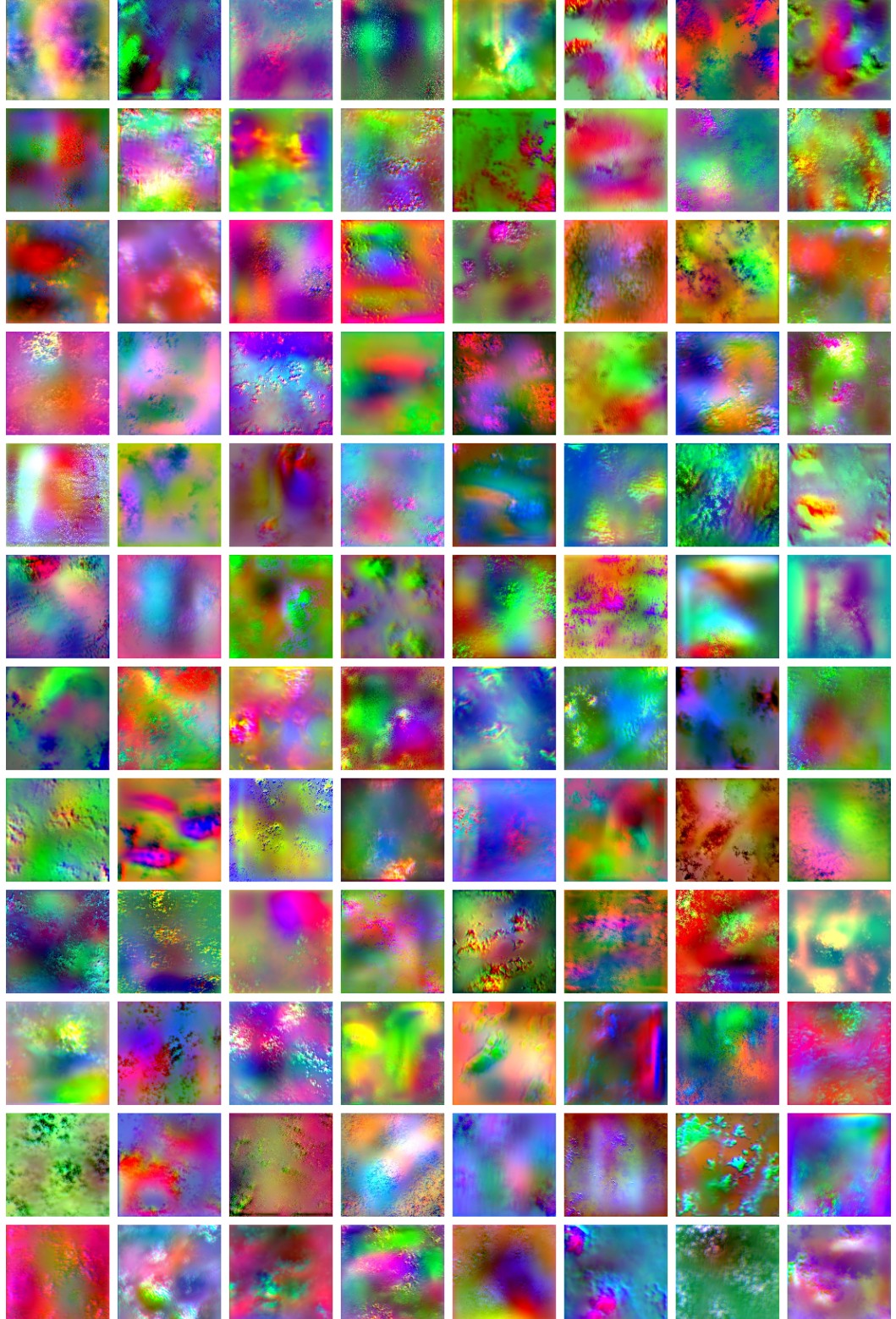

Figure 19: 96 random samples of the dataset o) StyleGAN - Sparse (letter as referenced in Fig. 2 of the main paper).

## S-3.16    StyleGAN - Oriented

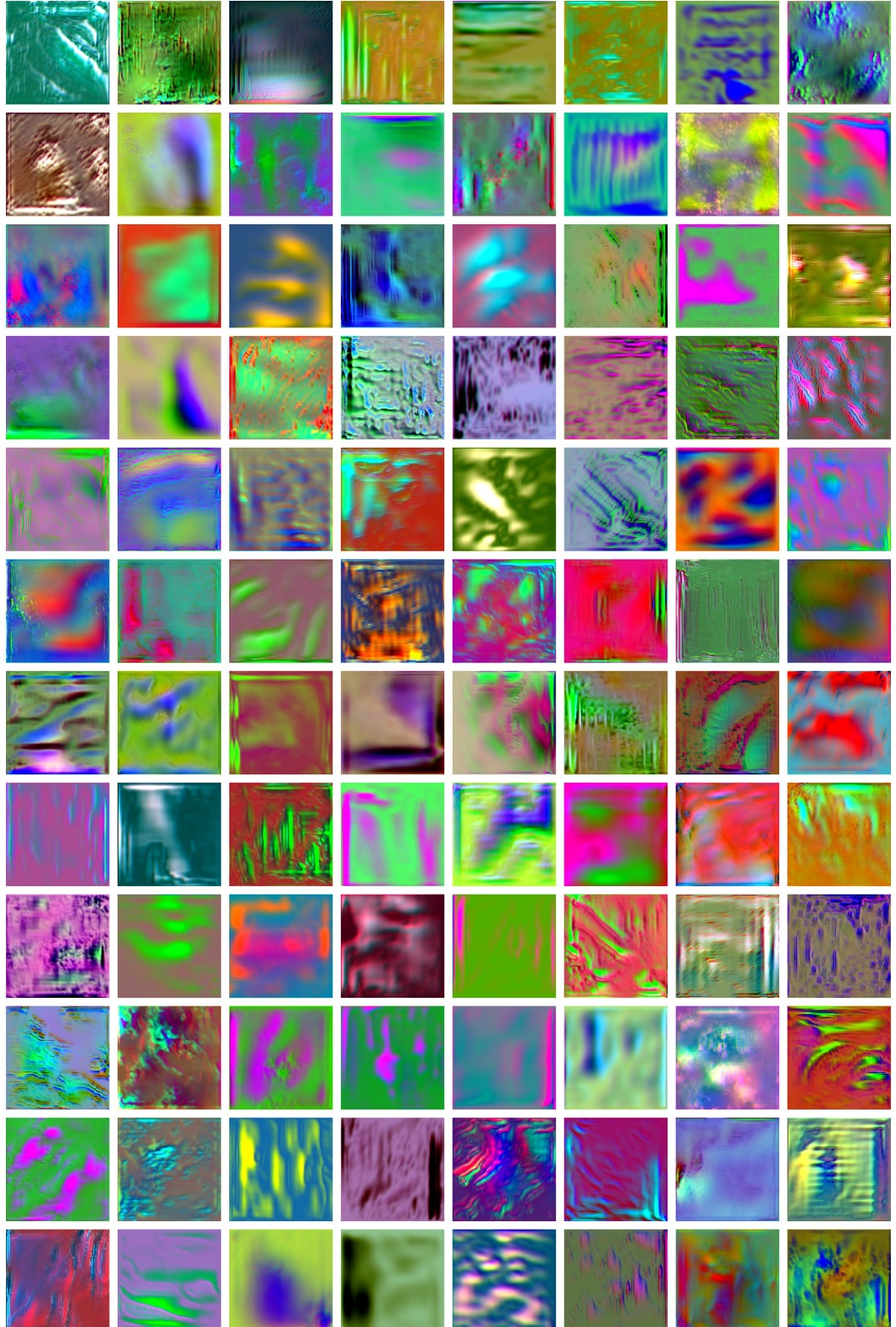

Figure 20: 96 random samples of the dataset p) StyleGAN - Oriented (letter as referenced in Fig. 2 of the main paper).

## S-3.17   Feature vis. - Random

Figure 21: 96 random samples of the dataset q) Feature vis. - Random (letter as referenced in Fig. 2 of the main paper).

**S-3.18    Feature vis. - Dead leaves**

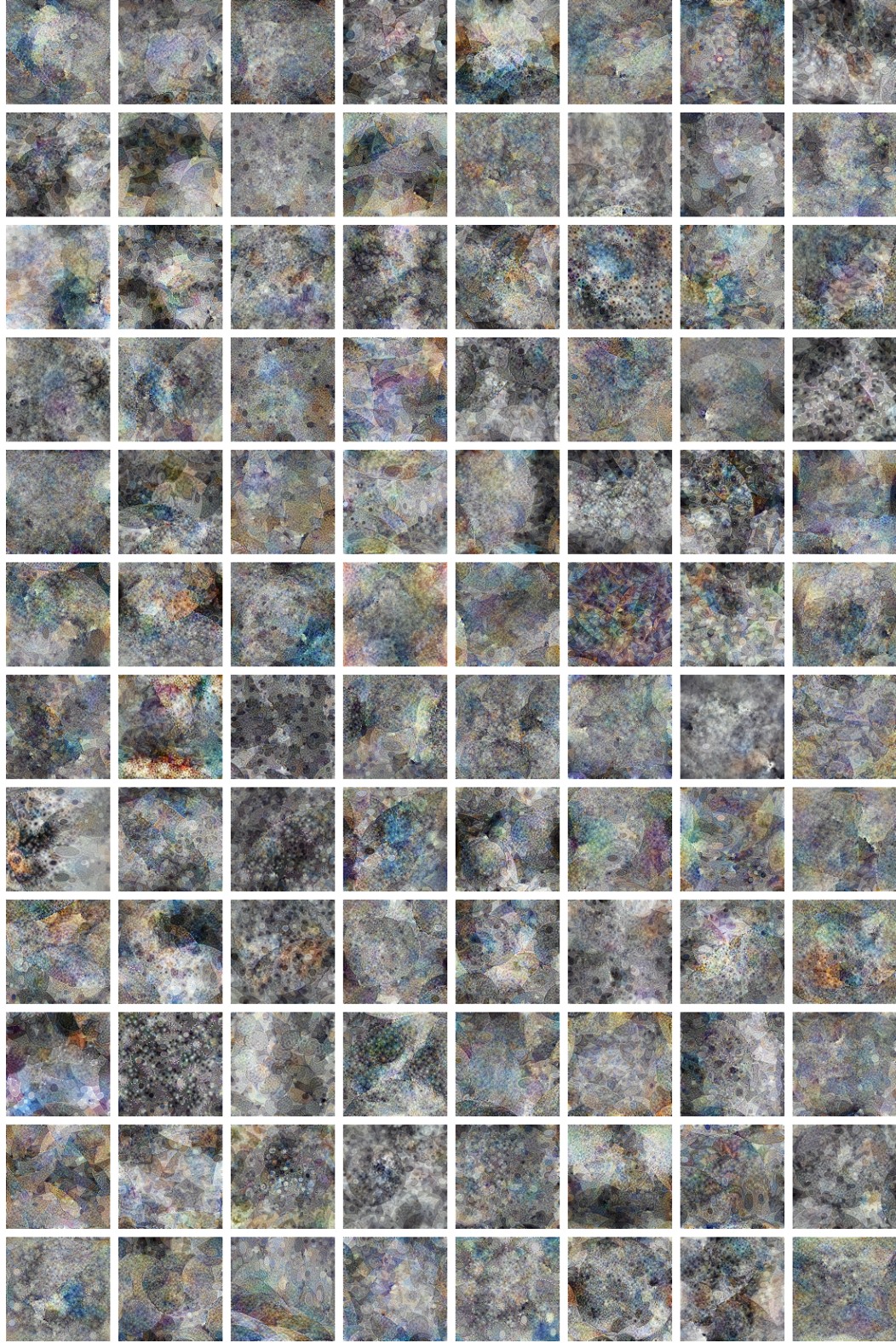

Figure 22: 96 random samples of the dataset r) Feature vis. - Dead leaves (letter as referenced in Fig. 2 of the main paper).