# OpenReview forum: "Learning to See by Looking at Noise"
_NeurIPS.cc/2021/Conference — NeurIPS 2021 Spotlight_

### Official Review · Reviewer_R4fm · 2021-07-09

**Rating:** 7
**Confidence:** 3

**Summary:**

The paper studies whether real natural images are necessary to train deep neural networks for computer vision applications. To that end, the authors create datasets from a large variety of synthetic image generation processes and evaluate the performance of neural networks trained using these synthetic images. In a comprehensive analysis the authors demonstrate that neural networks trained on synthetic images obtain surprisingly good results and also identify key properties of synthetic datasets that lead to good downstream performance.

**Limitations And Societal Impact:**

No apparent limitations and societal impact that is not addressed.

**Main Review:**

Strengths
As the authors correctly point out, recent advances in neural networks for visual recognition increasingly depend on larger and larger datasets. The apparent need for massive training sets raises many challenges as those training sets tend to be proprietary and data at this scale is incredibly difficult to obtain. As a consequence, methods that reduce the need for large datasets are of great interest to both practitioners as well as researchers. As such this paper is of great interest to the research community.

The analysis in this paper is outlined very clearly and the evaluation is comprehensive. Furthermore, the discussion and presentation of the results is focused and provides clear insights to the reader.

Weaknesses
One experiment that seems to be missing from the analysis are models that are trained on mixtures of datasets. In the paper each model is only trained on images generated from a single image generation process. As the authors point out, recall is very important for downstream performance. Could mixing multiple datasets lead to better coverage and thus downstream performance?

Minor comments:
Figure 7: For consistency, I suggest to also encode the accuracy of the baselines by color.


**Time Spent Reviewing:**

3

---

> ### Author Response · Authors · 2021-08-10
> **For Reviewer R4fm**
>
> We thank you for your insightful comments and suggestions.
>
> **1 - Mixtures of datasets**
>
> Please see the general comments for details and evaluation numbers; in summary, simply scaling the datasets up does not increase performance, but mixing datasets does.
>
> **2 - Minor comments**
>
> We will add accuracy colors to the baseline methods in Figures 7.b and 7.c.

---

> > ### Comment · Reviewer_R4fm · 2021-09-01
> > **Response**
> >
> > Thank you authors for your detailed responses.
> >
> > I think the paper provides some intriguing ideas and experiments and remain of the opinion that it is a good paper.

---

### Official Review · Reviewer_q39D · 2021-07-16

**Rating:** 6
**Confidence:** 4

**Summary:**

The paper investigates the effectiveness of synthetic data generated via procedural noise processes as training data for deep neural networks. Specifically, the authors consider fractals, computer graphics, dead leave models, statistical image models, and untrained GAN generators, as well as combinations thereof, as mechanisms to generate data, and train a network in self-supervised fashion on these different types of data. They find that deep networks trained in this fashion significantly outperform randomly initialized networks, and are competitive with deep networks that are trained on large natural image data sets on specialized domains (not necessarily natural images). The paper further presents a suite of experiments that aims to relate the properties of the synthetic data (e.g. color distribution) with the downstream performance of models trained on it.

**Limitations And Societal Impact:**

For limitations please see the main review. Societal impact is appropriately addressed.

**Main Review:**

Given the recent scaling trends of training deep networks on ever growing data sets, it is important to critically evaluate the benefits of large scale data. In that context the paper explores an interesting and highly relevant direction, questioning the necessity of large scale natural image data sets.

The paper investigates a broad and diverse choice of generative mechanisms, and also explores sound metrics in the systematic analysis of the relation between data set properties and downstream predictive performance. To my knowledge this selection of data sets and metrics has not been explored before.

The paper is well-written and mostly easy to follow.

The following points could be improved:
- How do the synthetic data sets compare to the single-image training from [26]? How does training on synthetic data compare to early self-supervised approaches (i.e. non-contrastive approaches) for example self-supervised learning via rotation prediction (Gidaris et al. ICLR’18)? I found it difficult to put the results in the paper into context, and it is therefore unclear to me how big the potential of the synthetic data approaches in the paper is.
- Are there ways to combine synthetic data with natural/real data, possibly with labels (e.g. similar to S4L (Zhai et al. CVPR’19))? Since there seems to be quite some gap to models trained on natural images and it is unlikely to deploy models purely trained using the data generation procedures from the papers, it would be interesting to see whether the synthetic data could be useful to augment natural/real data.
- Intuitively I would expect that networks trained on synthetic data do well on low-level texture based tasks and struggle more on tasks requiring high level visual features. An investigation into this aspect would make the paper more complete. Furthermore, for parameterized models it would be interesting to see how individual parameters affect downstream performance.

Minor comments:
- The different types of engineered filters described in Sec. 3.4. Maybe there is a way to present this more accessibly.
- Is the scale and cropping procedure (in particular the resolutions) described in L196 used throughout the paper? If so, is there an intuition why the patch size 64x64 is appropriate for training, assuming the testing resolution is higher?

Overall, I think the paper explores a highly relevant direction. However, I feel that the paper should be extended along the axes outlined above.


**Time Spent Reviewing:**

4

---

> ### Author Response · Authors · 2021-08-10
> **For Reviewer q39D**
>
> We thank you for your insightful comments and suggestions. We address your concerns in the following lines:
>
> **1 - Comparison to single image training**
>
> We tested training with a single image as in [1] by generating random crops uniformly and constructing a new dataset of 105k random crops. Using the small scale setting, the performance  is 0.13, 0.18 and 0.12 respectively, which is worse than using a randomly initialized CNN (0.20). This may be caused by the fact that the training approach we test may not be well suited for training with this dataset. For example, performance may increase if negative crops are selected within a minimum distance from positive pairs, but adapting the training algorithm to single image training is not in the scope of this paper.
>
> On the other hand, using the algorithm and architecture proposed in [1], the best performance reported by the authors on Imagenet1k is 0.334, which is worse than the best performance achieved with our large scale experiments on Imagenet1k (0.381).
>
> **2 - Focus on textures / low-level image structures**
>
> We agree with the reviewer’s intuition. However, a qualitative analysis using GradCAM shows that (compared to a ResNet50 trained on ImageNet1k) our model does not focus more on textures - in fact, the heatmaps produced by models pretrained on our random processes seem to be more localized. Since previous work [2] indicates that standard models often seem to base their judgement on textures and not on features like shapes, this is an interesting finding, and promising venue for further exploration. We will add these results to the paper and the supplemental material.
>
> **3 - Minor comments**
>
> The cropping resolution at 64x64 described in L196 corresponds only to the small scale experiments (Section 4.1). This corresponds to a random resized crop of size between 0.08 and 1 times the original resolution, selected uniformly at random (i.e. cropping a random patch from the original 128x128 image to the sampled size, and then rescaling the patch to 64x64 pixels). We apply this transformation for both the contrastive training and the training of the linear evaluation.
>
> At test time for the linear evaluation, we resize the original image to 70x70 pixels and then center crop a 64x64 patch. This was found to work well for this evaluation procedure in [3], as Imagenet images tend to have redundant information close to the borders.
>
> For the large scale experiments we use the default hyperparameters and augmentations as the original MoCo v2 implementation [4]. All These details will be included in the Supplementary Material as explained in the general comments.
>
> [1] Asano, Yuki M., et al. ‘A Critical Analysis of Self-Supervision, or What We Can Learn from a Single Image’. ArXiv:1904.13132 [Cs], Feb. 2020. arXiv.org, http://arxiv.org/abs/1904.13132.
>
> [2] Geirhos, Robert et al, “ImageNet-trained CNNs are biased towards texture; increasing shape bias improves accuracy and robustness”, ICLR 2019
>
> [3] Tongzhou Wang and Phillip Isola. Understanding contrastive representation learning through alignment and uniformity on the hypersphere. 119:9929–9939, 13–18 Jul 2020.
>
> [4] https://github.com/facebookresearch/moco

---

> > ### Comment · Reviewer_q39D · 2021-08-22
> > **Response**
> >
> > I thank the authors for their response. In particular, I think the observation that the proposed method clearly outperforms single image training and seems to be competitive with early self-supervised training methods, is encouraging.
> >
> > Based on the response, I decided to raise my score. Nevertheless, I would appreciate additional experiments combining the synthetic data with real (possibly labeled) data, hopefully showing improvements over standard training on real data, which would be immediately useful to the vision community.

---

### Official Review · Reviewer_96zt · 2021-07-16

**Rating:** 8
**Confidence:** 4

**Summary:**

The paper proposes to learn visual representations from procedurally generated images, drawing insights from natural image statistics. Non-trivial test accuracy was demonstrated on standard image classification datasets. The paper also considers what constitutes a good dataset for training wrt the statistics of individual images as well as the whole dataset highlighting precision-recall trade-offs when promoting naturalism vs diversity.

**Limitations And Societal Impact:**

~~Only one architecture and one objective function were considered.~~ The results would be more significant if they're shown to hold more generally. ~~The generation and evaluation, as well as image resolutions, were limited to a single setting.~~

**Main Review:**

Originality: good

Quality: good, could use more experiments

Clarity: good

Significance: good

Assessment:
===========
The approach is bold and follows nicely from classical and recent results on natural image statistics. The development and presentation are of the highest quality.  Good submission overall.

Requests and comments:
====================
- A justification for focusing on contrastive learning would be nice. It's not clear whether the conclusions generalize to other architectures and training regimes.
- Also, a more elaborate justification of the chosen baselines is needed.
- Unless I'm missing something, 105K images were used for training and 50K ImageNet images were chosen randomly for evaluation.  It would be nice to discuss the impact of gradually scaling up the size of the training set size, on both the test accuracy as well as the image/dataset statistics.
- I wonder if the proposed generative procedures can be intuitively scheduled into a more effective curriculum, rather than used all at once.
- The conclusions include an intriguing remark that synthetic data might prove better than pretraining with, e.g., ImageNet. It would be nice to consider tailoring the generation processes to push the performance on specific tasks, e.g., the structural categories in VTAB.
- It's natural to ask how many (fewer) real images would be needed to close the performance gap to an actual baseline, like a meta-learning twist?
- - By analogy to how mammalian vision is thought to be pretrained by retinal waves, it's likely that mammals still need even a few real examples and continue to get better, e.g., learning to read and even typoglycemia.
- - How does the precision-recall trade-off manifest, and how to think of naturalism vs diversity, in this (meta learning) paradigm?

**Time Spent Reviewing:**

3

---

> ### Author Response · Authors · 2021-08-10
> **For Reviewer 96zt**
>
> We thank you for your insightful comments and suggestions. We address your concerns in the following lines:
>
> **1 - Limitations**
>
> We test two (not only one) different protocols and models (ResNet50/MoCov2 and AlexNet/Lalign-Lunif) as described in Sections 4.1 and 4.2, with a wide range of datasets (Imagenet1k and VTAB, for a total of 20 datasets). The methods in Section 4.1 differ in architecture, training algorithm, loss function, input resolution and number of samples while still showing similar trends for the different generative processes analyzed.
>
>
> We will include more baselines and better justification for it, as explained in the general comments.
>
> **2 - Focus on contrastive learning**
>
> We focus on contrastive learning as this is the state of the art pretraining procedure given a set of unannotated images, or equivalently, an image model that one can sample from. For this form of unsupervised learning, we tested two different settings, as explained above.
>
> **3 - Scaling dataset size, mixing datasets and meta-learning**
>
> As mentioned in the general comments, mixing datasets does increase performance, while simply scaling the datasets up does not. We agree with the reviewer that studying a mixing curriculum and meta-learning strategies are indeed promising directions of future work, which however go beyond the scope of the current paper.

---

> > ### Comment · Reviewer_96zt · 2021-08-12
> > **Thanks for clarifying**
> >
> > I apologize I may have missed some of the details in Section 4. I'll update my review increasing the score.

---

### Official Review · Reviewer_tEQg · 2021-07-16

**Rating:** 6
**Confidence:** 4

**Summary:**

This paper provides a comprehensive study of using not naturally looking, synthetic images to train image feature extractors. Most of the synthetic images can be generated easily either in closed form or using a randomly initialized model. The obtained feature extractors result in significantly better models on natural images than randomly initialized counterparts after fine-tuning. Various types of synthetic images are evaluated in this paper to reveal the properties of images that can be used to train better feature extractors. In general, I feel this paper is studying a very interesting problem and a solution to the problem can be used in many practical scenarios where training data is difficult to collect or even not available, but the paper might need more technical contributions to be accepted. A potential way for improvement is to utilize the observations to generate better synthetic images from some random process that can improve the accuracy.

Strengths:
1. Evaluates a wide range of synthetic image types.
2. Comparing the statistical properties of synthetic and natural images, in an effort to uncover the important factors for good synthetic images.

Suggestions and Questions:
1. I feel the paper spends too many pages (3 pages) enumerating different image generation methods. These are important details, but maybe they should not take up 30%+ of the paper. The objective for training the models are ignored (I did not find it in the supplementary neither), though references to the paper of these methods are given. I feel it is important to clearly state the training objective in the main paper, and some details of image generation can be moved to the appendix.

2. I feel the observations in Sec. 5.1 are not conclusive. The accuracy do not seem to correlate well with the statistics considered. Maybe a more convincing way to verify the conclusions is to generate images that produce the stats that will likely produce highest accuracy according to the observations, and verify if it is the case. The correlations seem much stronger in Sec. 5.2, but the results are not surprising. For example, it is quite intuitive that if the training images have small FID to test images, thus more similar to test images, then the test accuracy will be higher.


## Summary after discussions

In general, I feel the findings in this paper is interesting and could potentially invoke investigations into the necessary conditions for the success of contrastive unsupervised learning for images, and would like to raise my score. However, I still encourage the authors to brand this paper as a scientific discovery rather than something that will have immediate impact on applications (addressing the privacy and bias concerns etc.). From the new results of mixing with real data, it seems adding images from random processes does not improve the results of only using real images. As to 3-Real unlabeled data, It also remains unclear whether pretraining with real images generalizes better than random images in the decentralized setting; it seems pretraining with real data will transfer better from the results of 4-Mixing with real data.


**Limitations And Societal Impact:**

Yes.

**Main Review:**

Yes

**Time Spent Reviewing:**

3

---

> ### Author Response · Authors · 2021-08-10
> **For Reviewer tEQg**
>
> We thank you for your insightful comments and suggestions. We address your concerns in the following lines:
>
> **1 - Experimental details**
>
> We will include further details about the training procedure and objective, as explained in the general comments. We see the main contribution of our paper as showing that images generated from simple properties and processes are sufficient to pretrain a classifier with surprisingly high accuracy. To this end, we believe it is necessary to describe the random processes in detail in order for the reader to understand the images properties that are captured by the individual processes.
>
> **2 - Statistics of data**
>
> The purpose of the analysis in Section 5.1 and 5.2 is to bring insights on what properties of the generated datasets correlate with performance (and which do not). Figure 6 together with the correlation coefficients (r=-0.57 and r=0.75) in L253 and L267 show that the color profile and image coherence correlate with performance. These are not sufficient conditions, as there may be datasets that perform badly though having a color profile similar to Imagenet. For example, a dataset consisting of solid color images with Imagenet color profile is likely to perform badly.
>
> Although using the insights in Section 5 to improve the random image generation is an interesting line of work, it exceeds the scope of this paper. This would either require a parameter sweep on the parameters that generate all the datasets (and performing the contrastive training for all of them) or devising a joint optimization scheme.
>
> Furthermore, one of the key contributions of our paper is to show the performance of models that have *never* been exposed to real data during the pretraining phase. Adjusting the noise processes in order to match the statistics of real images would again be a dependency on real data and might taint our datasets with dataset biases. We believe that it is in fact a strength of our method that this connection / matching of statistics does not exist.

---

> > ### Comment · Reviewer_tEQg · 2021-08-18
> > **Further questions**
> >
> > Thanks to the authors for their responses. I still do not feel my concerns are sufficiently addressed, and have some new questions after reading other reviews. My main concern is how significant these results are, and how much impact this paper will make to practice. In your comment, you first said that " We see the main contribution of our paper as showing that images generated from simple properties and processes are **sufficient** to pretrain a classifier with surprisingly high accuracy", but afterwards you said "...  that the color profile and image coherence correlate with performance. These are **not sufficient** conditions...", which seems a little bit contradictory though they are not referring to exactly the same thing. Is it possible to select a small set of real images according to these conditions that can produce better results than a random set of images? Another point is that it is never difficult to obtain unlabelled images for unsupervised learning. As another reviewer has asked, is there a way to use the findings from this paper to combine the random images with unlabelled real images to, e.g., obtain good results with only a small fraction of real images, or even find a criterion to reduce the sample complexity for training a good feature extractor? I will feel more confident in giving an accept if any of these questions are answered.

---

> > > ### Author Response · Authors · 2021-08-19
> > > **For Reviewer tEQg - Further questions**
> > >
> > > We thank you for your extra comments and suggestions. We address them next:
> > >
> > > **1-Significance of our results**
> > >
> > > The aim of our study is to explore a representation learning setting that has been seldomly explored before: using no real data. We believe that this is a relevant setting for the community as it tackles a fundamental question of computer vision: what is the minimal synthetic image model from which good general image representations can be learnt? Furthermore, the methods we propose have practical use cases (see **3-Real unlabeled data** below).
> > >
> > > With this, despite the image models proposed do not achieve the upper bound performance (i.e. training with real data as seen in Figure 3), the performance is far better than relevant previous work (38.12 vs 23.86 top-1 accuracy on Imagenet1k as seen in Table 1). Finally, we believe our study will encourage further exploration of this research direction.
> > >
> > > **2-Sufficient conditions**
> > >
> > > As you point out, our use of the term sufficient is misleading and we will rephrase the following sentence: *the experiments show that images generated from simple [properties and] processes are sufficient to pretrain a classifier with surprisingly high accuracy*. We will remove the text in brackets in the previous sentence to avoid confusion with the image properties analyzed in Section 5, which we show are *not sufficient* for high accuracy.
> > >
> > > **3-Real unlabeled data**
> > >
> > > Though it is usually easy to obtain unlabelled images for a known task, our method could be used in settings where the test distribution is not known beforehand or the training budget (once the task is known) is low. For example, it could be used for pretraining a model, deploying it at scale on mobile devices, and training the final linear layer on the edge (possibly for widely different tasks for each final node). Using this pretraining scheme with real data may raise copyright and bias concerns, which are mitigated when using random processes instead.
> > >
> > > **4-Mixing with real data**
> > >
> > > We agree that mixing real with synthetic data is an interesting direction to explore. We ran a pilot experiment mixing our Stylegan-oriented images with randomly sampled images from Places365, and obtained the following numbers:
> > >
> > > - 105k images StyleGAN-oriented only (as reported in the paper): 43.3%
> > >
> > > - 105k images StyleGAN-oriented + 1000 images from Places: 43.7%
> > >
> > > - 105k images StyleGAN-oriented + 105k images from Places: 46.6%
> > >
> > > - 105k images from Places only (as reported in the paper): 55.0%
> > >
> > > Using this protocol, we find that naively mixing datasets is not particularly effective; in particular, our synthetic dataset does not seem to allow us to use only a few real images and obtain similar accuracy as for the full real-image dataset. Mixing both the real and synthetic datasets underperforms the model trained with just real data, which may be caused by the training algorithm not being suited for these two completely different distributions. As was pointed out in the reviews, using a curriculum is a promising direction, but goes beyond the scope of this paper.

---

> > > > ### Comment · Reviewer_tEQg · 2021-08-22
> > > > **Comments on the updates**
> > > >
> > > > Thanks for the updates. In general, I feel the findings in this paper is interesting and could potentially invoke investigations into the necessary conditions for the success of contrastive unsupervised learning for images, and would like to raise my score. However, I still encourage the authors to brand this paper as a scientific discovery rather than something that will have immediate impact on applications (addressing the privacy and bias concerns etc.). From the new results of mixing with real data, it seems adding images from random processes does not improve the results of only using real images. As to **3-Real unlabeled data**, It also remains unclear whether pretraining with real images generalizes better than random images in the decentralized setting; it seems pretraining with real data will transfer better from the results of **4-Mixing with real data**.

---

### Author Response · Authors · 2021-08-10
**For all reviewers**

We thank the reviewers for their insightful suggestions and feedback. We appreciate that reviewers have found our manuscript to tackle an *interesting and highly relevant direction* and that the study is *focused and provides clear insights to the reader*. We will review the current version of the paper to include the following suggestions raised by multiple reviewers. Further suggestions and concerns raised by each of the reviewers are addressed in the comments specific to each reviewer. Where applicable, we will add the requested additional numbers and clarifications to the paper.

**1 - Explicit details of training setup**

We use the same hyperparameters that were found to work well in the original papers (which correspond to the default ones in their publicly available implementations). Though we cite these methods in our manuscript, we will include all hyperparameters explicitly in the Supplementary Material for completeness and easy reproducibility.

**2 - Improved baselines and training from a single image**

We will include an improved discussion of the baselines, moving the discussion of lower and upper bounds found in the caption of Figure 3 to the main text. We will also add an extra baseline consisting in training with crops extracted from a single image (using the images in [1], which show different degrees of complexity). The performance using a dataset consisting of 105k random crops for Images A, B and C in Figure 1 of [1], using the small scale setting is 0.13, 0.18 and 0.12 respectively, which is worse than using a randomly initialized CNN (0.20).


**3 - Mixing datasets and the impact of dataset size**

Several reviewers mentioned the potential of mixing the outputs of different noise processes to increase performance. We find that indeed mixing datasets has some benefit. In the small-scale experiment, using the best performing method from each noise process category (StyleGAN-oriented, Dead Leaves-Textures, Feature Vis-Dead Leaves, Spectrum+Color+WMM) results in an increase of accuracy of about 1% when keeping the size of the training set constant at 105K images (i.e. we use 25% of each dataset). Combining the full datasets, thereby increasing the size of the training set to 420K images yields another 1% accuracy increase (ImageNet100 Acc@1=45.64%).
However, simply adding more images of the noise processes does not significantly increase accuracy (StyleGAN-oriented: Acc@1=43.28% with 105K images, Acc@1=43.5% with 420K images). We will add all these results to the paper.

[1] Asano, Yuki M., et al. ‘A Critical Analysis of Self-Supervision, or What We Can Learn from a Single Image’. ArXiv:1904.13132 [Cs], Feb. 2020. arXiv.org, http://arxiv.org/abs/1904.13132.

---

### Decision · Program_Chairs · 2021-09-27

**Decision:**

Accept (Spotlight)

**Comment:**

The paper addresses an under-explored research question related to the necessity of large datasets and the promise of doing representation learning pretraining using only noisy processes. The reviewers found the topic as well as the comprehensive study very interesting and great area of study. I concur in that I believe the paper could help foster more research in this direction, so I believe it is a great candidate for a spotlight. However, there are some lingering requests from the reviewers: I strongly advise the authors to follow up on these.